# Analysis of gene network bifurcation during optic cup morphogenesis in zebrafish

Lorena Buono [1,2], Jorge Corbacho [1], Silvia Naranjo [1], María Almuedo-Castillo [1], Tania Moreno-Marmol [2], Berta de la Cerda [3], Estefanía Sanabria-Reinoso [1], Rocío Polvillo [1], Francisco-Javier Díaz-Corrales [3], Ozren Bogdanovic [4,5], Paola Bovolenta [2,6 ✉] & Juan-Ramón Martínez-Morales [1 ✉]

Sight depends on the tight cooperation between photoreceptors and pigmented cells, which derive from common progenitors through the bifurcation of a single gene regulatory network into the neural retina (NR) and retinal-pigmented epithelium (RPE) programs. Although genetic studies have identified upstream nodes controlling these networks, their regulatory logic remains poorly investigated. Here, we characterize transcriptome dynamics and chromatin accessibility in segregating NR/RPE populations in zebrafish. We analyze cis-regulatory modules and enriched transcription factor motives to show extensive network redundancy and context-dependent activity. We identify downstream targets, highlighting an early recruitment of desmosomal genes in the flattening RPE and revealing Tead factors as upstream regulators. We investigate the RPE specification network dynamics to uncover an unexpected sequence of transcription factors recruitment, which is conserved in humans. This systematic interrogation of the NR/RPE bifurcation should improve both genetic counseling for eye disorders and hiPSCs-to-RPE differentiation protocols for cell-replacement therapies in degenerative diseases.

[1] Centro Andaluz de Biología del Desarrollo-CABD (CSIC/UPO/JA), Seville, Spain. [2] Centro de Biología Molecular Severo Ochoa (CSIC/UAM), Seville, Spain. [3] Centro Andaluz de Biología Molecular y Medicina Regenerativa-CABIMER (CSIC/US/UPO/JA), Seville, Spain. [4] Genomics and Epigenetics Division, Garvan Institute of Medical Research, Sydney, NSW, Australia. [5] School of Biotechnology and Biomolecular Sciences, University of New South Wales, Sydney, NSW, Australia. [6] CIBERER, ISCIII, Madrid, Spain. ✉email: pbovolenta@cbm.csic.es; jrmarmor@upo.es

Vision depends on the close physical and physiological interaction between pigmented cells and photoreceptors. Pigmented cells protect photoreceptors by maintaining their homeostasis through growth factors secretion, visual pigments recycling, and outer segments phagocytosis[1]. Fate map experiments in vertebrates have shown that the precursors of the neural retina (NR) and retinal pigmented epithelium (RPE) derive from the undifferentiated eye field[2]. In the permanently growing retinas of teleost fish, stem cells located at the ciliary marginal zone are capable of producing both cell types[3]. Furthermore, in tetrapods RPE precursors retain a certain potential to acquire NR identity[4] and vice versa[5].

In zebrafish, the RPE and NR presumptive domains start differentiating at the optic vesicle stage from the medial (ML) and lateral (LL) epithelial layers, respectively[2,6]. The specification of both retinal domains occurs simultaneously to the folding of the vesicle into a bi-layered cup and entails profound cell shape changes in each domain. Precursors at the LL elongate along their apicobasal axis, differentiate as NR, and constrict basally to direct the folding of the retinal neuroepithelium[7,8]. In contrast, precursors at the ML either acquire a squamous epithelial shape and differentiate as RPE, or flow into the LL to contribute to the NR domain[9–12]. In other vertebrates, the specification of the NR/RPE domains and the morphogenesis of the optic cup follow a pattern similar to that of the zebrafish, though the relative weight of the different morphogenetic mechanisms may vary among species[13].

During the last decades, numerous studies have investigated how the RPE and NR domains get genetically specified[14]. Eye identity is established in the anterior neural plate by the early activation of a gene regulatory network (GRN) pivoting on a few key regulators collectively known as eye field transcription factors (EFTF): including Lhx2, Otx2, Pax6, Rx, or Six3[15]. Upon the influence of inductive signals derived from the lens epithelium (FGFs) or the extraocular mesenchyme (Wnts and BMPs), the eye field network branches into the mutually exclusive developmental programs of the NR and RPE[14,16]. Classical genetic experiments performed mainly in mice have identified key nodes of the NR and RPE specification networks. The TF-encoding gene *Vsx2* (previously known as *Chx10*) was identified as the earliest determination gene differentially expressed in NR but not in RPE precursors[17]. *Vxs2* plays an essential role in specifying the NR domain by restraining the RPE identity through direct repression of the TF Mitf[5,18–20]. Other homeobox regulators, many inherited from the eye field specification network, contribute to the network of NR specifiers. This list includes *Lhx2, Sox2, Rx, Six3*, and *Six6* genes, which are required either for NR maintenance or for suppressing RPE identity[21,22]. The establishment of the RPE network depends instead on the cooperative activity between Mitf and Otx factors[23,24]. Classical experiments in mice showed that inactivation of either *Mitf* or *Otx* genes results in an RPE differentiation failure, with the alternative acquisition of NR molecular and morphological features[25–27]. More recently, mutation of the main effectors of the Hippo pathway, *Yap* and *Taz*, showed the essential role of these co-regulators in the differentiation of the RPE lineage[28,29].

Despite the identification of these key regulators, the architecture of the NR and RPE specification subnetworks is far from being well understood[30]. Systematic attempts to reconstruct retinal GRNs using next-generation sequencing (NGS) methods have focused mainly on the differentiation of neuronal types at later stages of development or in epigenetic changes linked to retinal degeneration[31–33]. Very recently, scRNA-seq has been proved a useful tool to characterize cell heterogeneity and infer differentiation trajectories in human tissues and retinal organoids[34,35]. However, a comprehensive analysis of the NR and RPE bifurcating networks would require a deeper sequencing coverage approach, while dealing with the limited cell population size at optic cup stages.

In this study, we address previous limitations by applying a combination of RNA-seq and ATAC-seq on sorted NR and RPE populations. Using the zebrafish retina as a model system, we follow transcriptome dynamics and chromatin accessibility changes in both cell populations, as they depart from a common pool of progenitors. The cross-correlation of RNA-seq and ATAC-seq data allow us to identify activating and repressing cis-regulatory elements (CREs), discover enriched transcription factor (TF) binding motifs, and unveil relevant downstream targets for the main specifiers. Our analyzes confirm previously known TFs as central nodes of the eye GRNs, and more importantly, provide information on their recruitment sequence both in zebrafish and human cells.

## Results

### Analysis of specification networks in isolated NR and RPE precursors.
To examine the bifurcation of the regulatory networks specifying the NR and RPE domains in zebrafish, we focused on the developmental window comprised between stages 16 hpf and 23 hpf (Fig. 1a). Within this period, the optic vesicle transits from a flattened disk in which the lateral and medial layers are still similar in terms of cell shape and volume, to an optic cup stage in which the NR and RPE cells are morphologically differentiated: bottle-shaped for retinal neuroblasts and flat for RPE precursors[2]. It is important to state that at the end of this window (23 hpf), neurogenesis has not started yet in the retina, neither the RPE display pigmentation. We used the transgenic lines *E1_bHLHe40: GFP* and *vsx2.2:GFP* to isolate the NR and RPE populations at 18 and 23 hpf by FACS (see "Methods"; Fig. 1b). Both *vsx2* and *bhlhe40* are among the earliest markers reported in zebrafish for the NR and RPE domains respectively[36–38]. To isolate early precursors at stage 16 hpf we took advantage of the fact that the *vsx2.2:GFP* transgene is transiently expressed in all progenitors and gets restricted to the NR as progenitors of the medial layer start expressing *bHLHe40*[8,36]. Populations isolated by flow cytometry were analyzed by RNA-seq, at all stages, and by ATAC-seq at 23 hpf (Fig. 1a).

The transcriptomic analysis of the different cell populations highlighted thousands of differentially expressed genes (DEGs) throughout development and between domains (Supplementary Fig. 1a, b; Supplementary Data 1). A multidimensional analysis of these changes revealed that an extensive divergence between the NR and RPE transcriptomes already occurs within the first 2 h (16–18 hpf) of optic vesicle folding (Fig. 1c). Later on, between 18 hpf and 23 hpf, the transcriptome divergence between the NR and RPE progresses, although is more modest within each domain (Fig. 1c and Supplementary Fig 1b).

### Chromatin accessibility in NR vs. RPE precursors identifies cis-regulatory modules.
To gain further insight into the architecture of the optic cup GRNs, we sought to identify differentially open chromatin regions (DOCRs) in the morphologically divergent NR and RPE domains. To this end, we performed ATAC-seq experiments from isolated NR and RPE precursors at 23 hpf. A total of 238,369 open chromatin regions (OCRs) were detected using this approach. After statistical analysis, a fraction of the peaks were identified as DOCRs, corresponding to putative CREs that are more active in the NR than in the RPE or vice versa. Approximately 12.6% of all peaks (30172 peaks) were differentially open with an adjusted $p$-value < 0.05. This proportion was reduced to 4.8% when the adjusted $p$-value was lowered to <0.001 (Supplementary Fig. 2a, b; Supplementary Data 2). Regardless of the adjusted $p$-value, we observed a larger number of DOCRs in

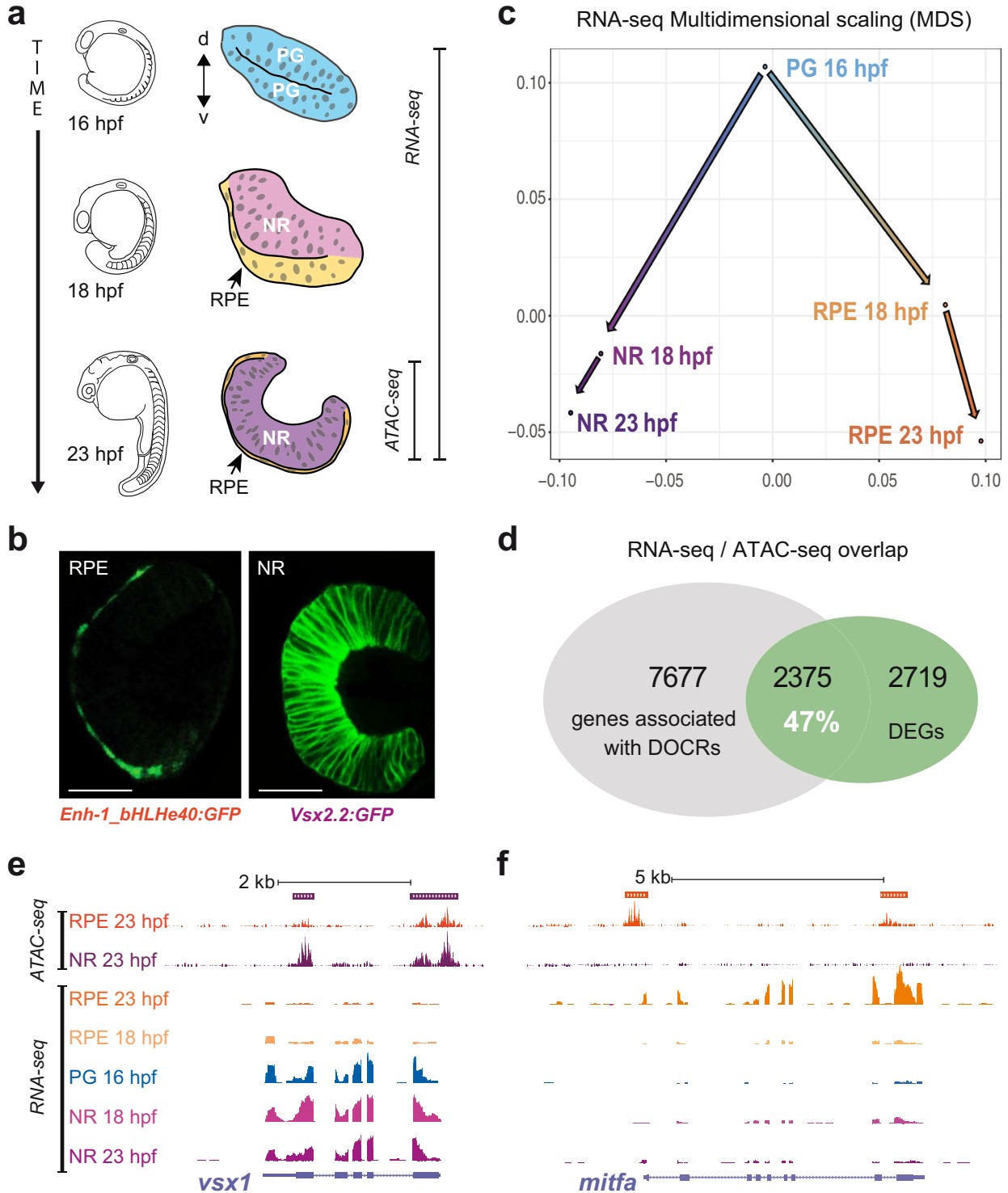

**Fig. 1 Experimental setup and raw data. a** Schematic representation of optic vesicle morphogenesis from undifferentiated retinal progenitor cells (PG, blue), at 16 hpf, to segregation and differentiation of the neural retina (NR, purple) and pigmented epithelium (RPE, orange) domains at 18 and 23 hpf. NR and RPE populations isolated by flow cytometry were analyzed by RNA-seq and ATAC-seq at the stages indicated. **b** Stage 23 hpf confocal sections from the zebrafish transgenic lines used to mark and isolate the RPE (*E1_bHLHe40:GFP*) and NR (*vsx2.2:GFP*) populations. Bar = 50 μm. **c** Multidimensional scaling analysis of the RNA-seq data showing a progressive transcriptomic divergence between the NR and RPE domains. **d** Percentage of overlapping between differentially expressed genes (DEGs) and genes associated with differentially opened chromatin regions (DOCRs). **e, f** Overview of ATAC-seq and RNA-seq tracks (UCSC browser) for representative NR (*vsx1*; (**e**)) and RPE (*mitfa*; (**f**)) specific genes. Solid bars on the top indicate DOCRs. If purple, the DOCR is more accessible in the NR. On the contrary, if orange, the DOCR is more accessible in the RPE. In the depicted case all the DOCRs are accompanied by increased transcription of the associated gene in the corresponding tissue.

the RPE than in the NR (18,909 vs. 11,263; adjusted $p$-value < 0.05), and their average chromatin accessibility fold change was higher than that of NR associated DOCRs (Supplementary Fig 2b, c). Analysis of the distribution of OCRs in the genome showed no evident differences between NR and RPE for the whole set of OCRs. However, there was a difference in average localization between the entire OCRs and the subsets of DOCRs in the genome. When compared to the distribution of all the OCRs a noticeable depletion of DOCRs located near the promoter regions was evident in both the NR and RPE domains. The proportion of NR peaks near the promoter was 11.91% for OCRs vs. 4.6% for DOCRs; whereas for the RPE peaks the proportion was 10.03% for OCRs vs. 3.09% for DOCRs (Supplementary Fig. 2d), showing that domain-specific cis-regulatory modules tend to occupy more distal positions in the genome. Gene ontology analysis of terms from the category Biological Process enriched in the list of genes associated with DOCRs yields results consistent with the analyzed tissue. Genes associated with NR DOCRs were enriched in terms related to nervous system development, neuron differentiation, and eye morphogenesis, whereas genes associated with RPE DOCRs were enriched in terms such as melanocyte differentiation and epithelial differentiation (Supplementary Fig. 3).

To further characterize the NR and RPE-specific DOCRs, we next interrogated their DNA methylome profiles. We examined base-resolution DNA methylome (WGBS) and hydroxymethylome (TAB-seq) data of zebrafish embryogenesis and adult tissues[39], to obtain insight into DOCRs DNA methylation (5mC) dynamics. Supervised clustering ($k = 2$) of 5mC patterns separates both datasets into two well-defined clusters (Supplementary Fig. 4a, b). The first consists of distal regulatory regions that initiate active demethylation at ~24 hpf, as demonstrated by strong 5-hydroxymethylcytosine (5hmC) enrichment and the progressive loss of 5mC starting at 24 hpf. Importantly, this cluster is highly enriched in H3K27ac, an active enhancer mark, and depleted of H3K4me3, which indicates their identity as bona fide enhancers[40]. The second cluster of both NR and RPE datasets is hypomethylated, depleted of hmC, and enriched in the promoter mark H3K4me3, in line with a CpG island promoter character. To test whether active, tet-protein dependent DNA demethylation is required for the establishment of open chromatin structure at NR and RPE ATAC-seq peaks, we interrogated chromatin accessibility in wild type and triple tet (tet1/2/3) morphants in whole 24 hpf embryos[39]. Data from two independent MO knockdown experiments revealed that their chromatin accessibility signature is significantly (Mann–Whitney–Wilcoxon Test, $P < 2.2e-16$) reduced for NR ATAC-seq peaks in tet morphants when compared to wild type embryos, even when whole embryos were assessed (Supplementary Fig. 4c). This decrease was not visible in RPE peaks, likely because RPE enhancers are active in a much smaller cell population and their chromatin mark is masked by the en masse approach (Supplementary Fig. 4d).

**Cross-correlation of RNA-seq and ATAC-seq data reveals activating and repressing CREs.** Integration of ATAC-seq and RNA-seq data has been used as a powerful tool to explore the architecture of developmental GRNs[33,41]. By intersecting RNA-seq and ATAC-seq datasets, we observed a substantial overlap between DEGs and genes associated with DOCRs at 23 hpf. Thus, 47% of all DEGs are associated with at least one DOCR (Fig. 1d). Specific examples of ATAC-seq and RNA-seq tracks are shown for the *vsx1* and *mitfa* loci, NR, and RPE respective markers (Fig. 1e, f). The cross-analysis of ATAC-seq and RNA-seq data allowed us to classify DOCRs (putative CREs) in two groups: those correlating with upregulated genes, here termed "activating CREs" and those correlating with silenced genes, here termed

"repressing CREs" (Fig. 2a, b). Illustrative examples of activating and repressing peaks are provided for the *six3a* and *otx2* loci (Supplementary Fig. 5a, b). When their average distance to the closest transcription start site (TSS) was examined, we observed a significant trend: both for NR and RPE peaks, activating CREs tend to occupy positions closer to the TSS than the repressing regions (Supplementary Fig. 5c). A possible explanation for this difference is the positive bias introduced by proximal promoters, which are typically enriched in binding sites for activators[42]. To gain insight into the regulatory logic of the NR and RPE domains, we calculated the number of associated activating and repressing CREs for both the whole DEGs and the subset of differentially expressed TFs. Notably, as expected from their regulatory complexity, the average number of CREs per gene (either activating or repressing) was significantly higher for TFs than for a control sample of randomized DEGs (Chi-square Test; $P < 0.0001$) (Fig. 2c). More importantly, the NR TFs associated with activating CREs outnumbered TFs associated with repressing regions (182 vs. 69 respectively), whereas the opposite was observed in the RPE. Indeed, in this tissue, we detected a robust repressive cis-regulatory logic, with 173 TFs associated with at least one repressing CREs and only 110 associated with activating ones (Fig. 2b, c). This opposite trend, better appreciated in histogram graphs, suggests that the NR specification program is sustained mainly by the activation of transcriptional regulators, whereas RPE determination seems to require primarily the repression of the NR-specific-TFs (Fig. 2d, e).

**Distinctive sets of TFs and cytoskeletal components are progressively recruited during NR and RPE specification.** To efficiently analyze expression dynamics, as the NR and RPE networks bifurcate, we used a gene clustering approach. We applied both hierarchical and partitioning soft clustering for the classification of genes encoding for TFs or cytoskeletal components. We focused on these two gene categories to examine not only transcriptional lineage specifiers but also terminal effectors operating in the divergent cell shape changes observed between the NR and RPE domains. Using partitioning soft clustering we established 25 groups of gene expression variation from progenitors towards NR or RPE, for both TFs and cytoskeleton components (Supplementary Fig. 6a, b; Supplementary Data 3 and 4). This approach identified several expression clusters, which are distinctive for each domain and developmental stage: e.g., clusters 1, 13, and 23 for 16 hpf progenitors in the TFs category (Supplementary Fig. 6a); or 6, 11, and 21 for 23 hpf RPE precursors in the cytoskeletal components' analysis (Supplementary Fig. 6b). Using a hierarchical clustering approach, we could aggregate the small sub-groups into six main clusters for TFs and eight for cytoskeletal components, all of them linked mainly to a specific domain and/or developmental stage (Fig. 3 and Supplementary Fig. 7; Supplementary Data 5 and 6). To define precise time windows of expression and to infer regulators order of action, we focused our attention on the identity of TFs belonging to the different large clusters. Significant TFs in cluster #5, corresponding to 16 hpf progenitors, include *rx3*, an early eye specifier with a known role in optic vesicle evagination[43–45]; and *her* factors required to maintain the progenitor neural state[46] (Fig. 3b). As expected, Cluster #1, corresponding to NR 23 hpf precursors, contains many of the acknowledged retinal specifiers, such as *rx1*, *rx2*, *sox2*, *six3a*, *six3b*, *six6b*, *vsx1*, *vsx2*, *hmx1*, *hmx4*, and *lhx2b*[14], which already increased their expression at 18 hpf (Fig. 3b). Surprisingly, known RPE specification genes such as *otx2* and *mitfa*[47], included in cluster #4, do not increase their levels significantly in RPE precursors at 18 hpf, peaking only later at 23 hpf (Fig. 3b). In contrast, TFs included in cluster #3, such as

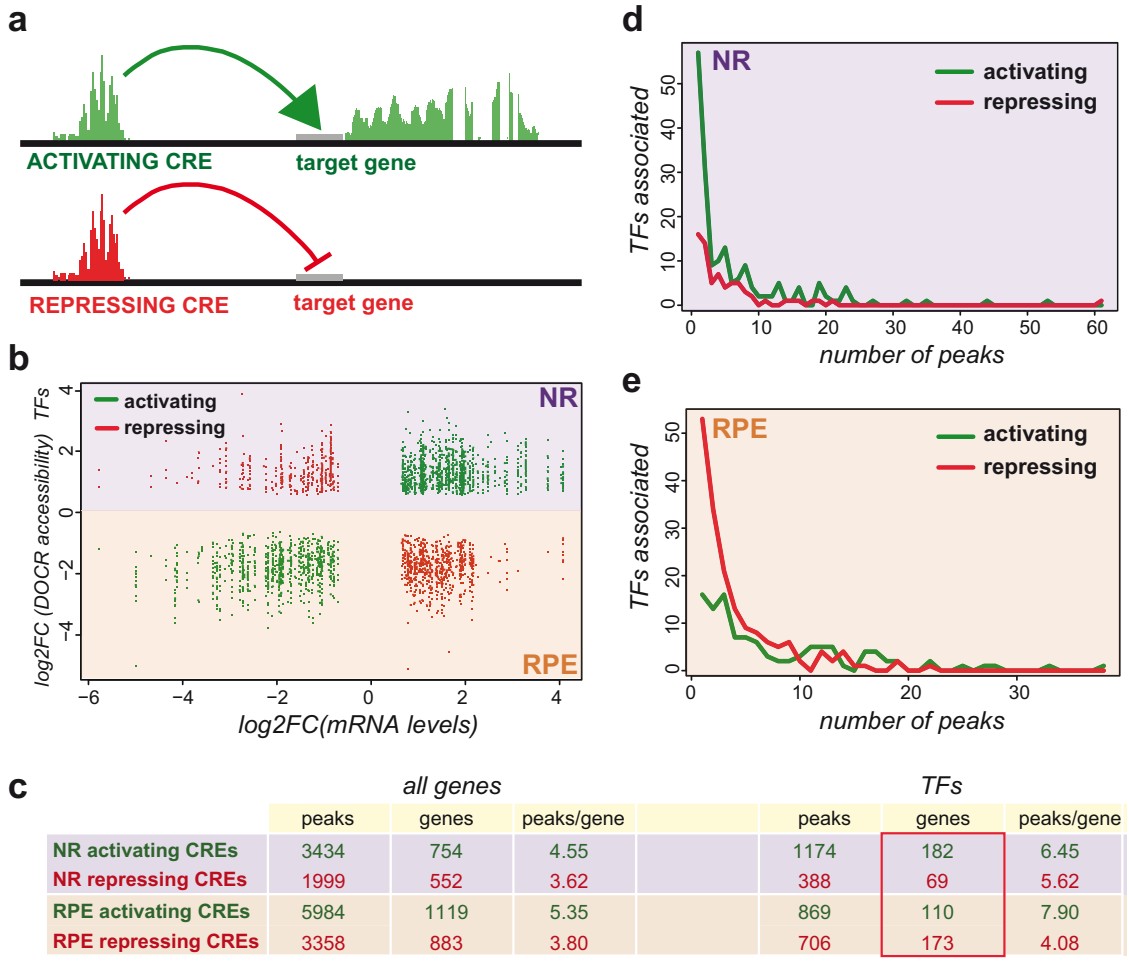

**Fig. 2 CRE configuration in the NR and RPE domains. a** Schematic representation of the functional relationship between DOCRs, either activating (green) or repressing (red), and their associated DEGs. **b** The graph illustrates the correlation between the levels of differentially expressed TFs (log2FC) and the accessibility of their associated DOCRs (log2). **c** The table summarizes the number of activating or repressing CREs associated with either all the DEGs or only with the differentially expressed TFs. **d**, **e** Histograms correlating the number of TFs associated with activating or repressing CREs to the number of peaks per gene, for both the NR (**d**) and RPE (**e**) domains.

*tead3b*, *tfap2a* and *tfap2c* started to rise, when not peaking, in the RPE at 18 hpf. Notably, some of these TFs, including *tcf12*, *smad6b*, and especially *vgll2b* not only peak at 18 hpf, but also rapidly decrease at 23 hpf (Fig. 3b). This observation suggests the existence of two waves of TFs regulating the identity of the RPE domain.

Next, we focused on the content of the hierarchical clusters associated to cytoskeletal components (Supplementary Fig. 7a). While the analysis of the cytoskeletal genes upregulated in PG returned no significant enrichment for terms annotated in the Cellular Components, a similar analysis for cytoskeletal genes in the NR clusters yielded significantly enriched terms associated with microtubules and centrosomes (Supplementary Fig. 7b). This observation is in line with the elongation of the apicobasal axis and the polarization of the microtubules reported for NR precursors in zebrafish[7]. In contrast, the only enriched term in the RPE cytoskeletal clusters was "intermediate filaments" (Supplementary Fig. 7b), consistent with the RPE acquiring a squamous epithelial character.

**Motif enrichment analysis suggests redundancy and cooperativity in the NR and RPE networks.** To further explore the regulatory logic of the NR and RPE gene networks, we investigated overrepresented motifs within the DOCRs associated with

each domain. Motif enrichment analysis of the differentially open regions in the NR identified a highly significant over-representation of the homeobox and sox TF binding motifs (Fig. 4a; Supplementary Data 7). The core homeobox binding motif (5′–TAATT–3′) is shared by TFs from the homeodomain K50 PRD-class; such as *vsx1*, *vsx2*, *rx1*, *rx2*, and *rx3*; LIM-class, such as *lhx2b*; and NKL class; such as *hmx1* and *hmx4* (Fig. 4a). All of them are well-known retinal specifiers contained in group #1 of our hierarchical clustering analysis (NR 23 hpf). As they are co-expressed in the retina, it is likely that they cooperate to target a partially overlapping set of cis-regulatory modules. To explore this possibility, we retrieved the individual position weight matrixes (PMW) associated with both homeobox and sox TFs from available databases and used this information to explore their functional synergy in the retinal DOCRs (see "Methods"). Two different approaches were used: (i) calculating the co-occurrence rate of binding sites for two different TFs in the same CRE (co-occupancy); and (ii) estimating the percentage of binding sites for TFs located in different CREs but still associated with the same gene (co-regulation) (Fig. 4b, c). These analyzes predicted a high degree of interconnectivity within the NR network. Thus, the average co-occurrence rate of two different theoretical binding sites in the same peak (co-occupancy) was 25.6%. This combinatorial activity becomes more pronounced when

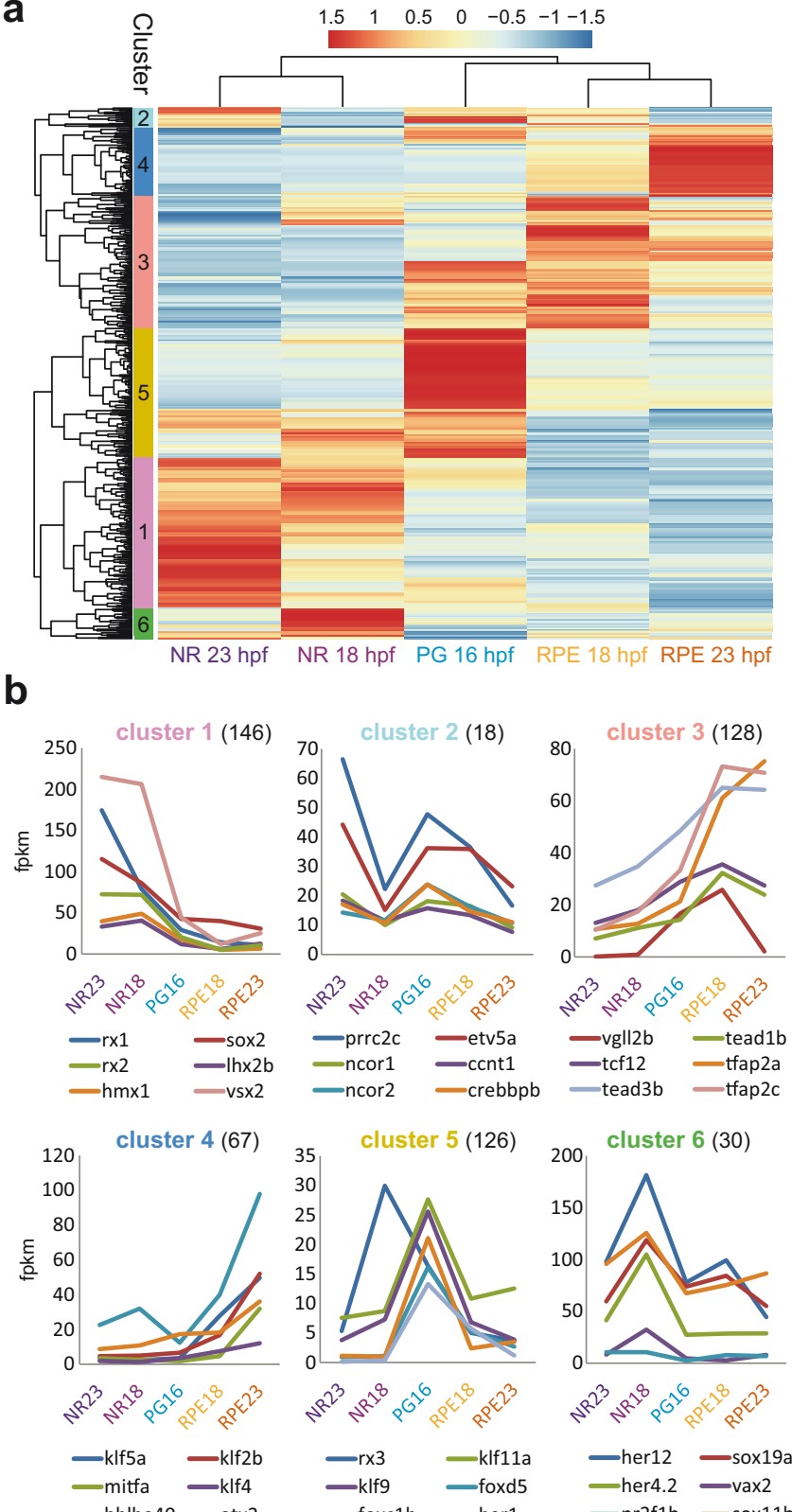

**Fig. 3 Hierarchical clustering of TFs gene expression variations during optic cup development. a** Hierarchical clustering output shows TFs expression trends in the distinct domains and stages. Gene expression values, normalized by row, are indicated with a red to blue graded color. Note that most clusters comprise a particular domain and developmental stage. **b** Relative transcript level changes for significant TFs (*n*) within each cluster.

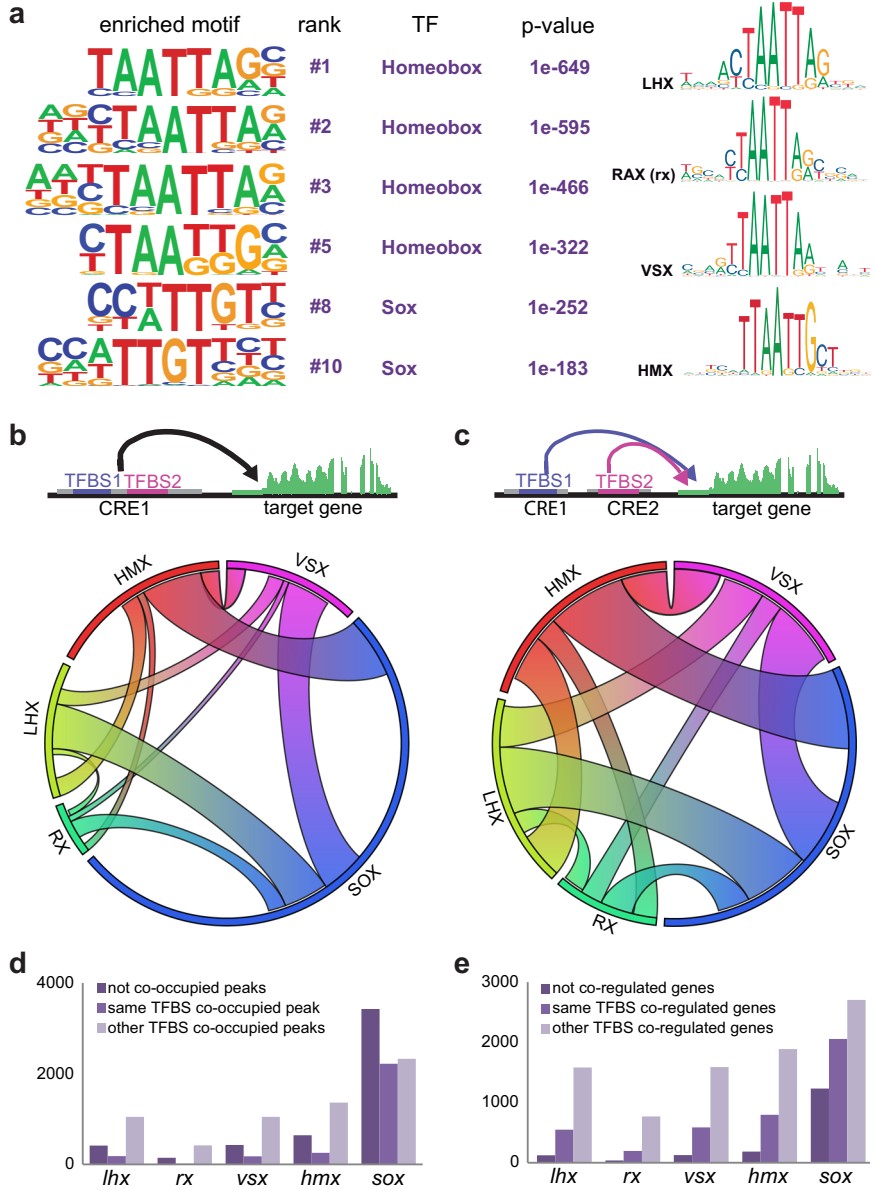

**Fig. 4 NR motif enrichment analysis. a** Representative TF binding motifs enriched in NR DOCRs as identified by HOMER. The binding motif similarity among neural retina TFs of the homeobox family is indicated (http://jaspar.genereg.net). **b** Circoplot illustrating the co-occupancy rate of TFBS in the same DOCRs for the main TFs identified in the motif enrichment analysis. **c** Circoplot illustrating the degree of co-regulation between TFs regulating the same gene through different DOCRs. **d** Number of CREs containing the main TFs identified in the motif enrichment analysis classified according to their co-occupancy. **e** Number of genes associated with CREs containing the main TFs identified in the motif enrichment analysis, classified according to their co-regulation.

examining the binding sites in different CREs associated with the same gene (co-regulation), which was in average 46.6%. Indeed, when individual TF binding sites were investigated, the number of co-occupied peaks or co-regulated genes was in general much larger than those instances in which the binding site was found isolated (Fig. 4d, e). Even when these are predicted binding sites, our analyzes suggest a strong cooperative activity of homeobox and sox factors in driving the NR developmental program.

In the RPE, a motif enrichment analysis of the specific DOCRs also revealed a number of binding motifs that correspond to the TFs identified as up-regulated in the RNA-seq analysis (Fig. 5a; Supplementary Data 8). We found very significant enrichment for tfap2a and tfap2c binding motifs, in agreement with the early expression of these factors in the RPE. This considerable overrepresentation points to a prominent position of these factors

within the regulatory hierarchy, which is in line with their role in the specification of the pigmented tissue[48]. In addition, a significant enrichment for bHLH, tead, and otx2 binding motifs were also observed. Importantly, the core bHLH motif (5′–CACGTG–3′) is shared by members of the family expressed in RPE, such as mitfa, bHLHE40, bHLHE41, and tfec (Fig. 5a). As already investigated for the NR TFs, we examined the potential cooperation among these RPE specifiers (Fig. 5b–e). Interestingly, whereas the average rate of co-regulated genes in the RPE did not differ much from that found in NR (48% vs. 46.6%; Fig. 5e), the rate of co-occupied peaks dropped from 25.6% in the NR to 11.7% in the RPE (Fig. 5d). This observation suggests that the RPE network is less dependent on TFs cooperativity within the same CRE than the NR network. In such a regulatory scenario, each TF may trigger transcriptional sub-programs within a broader

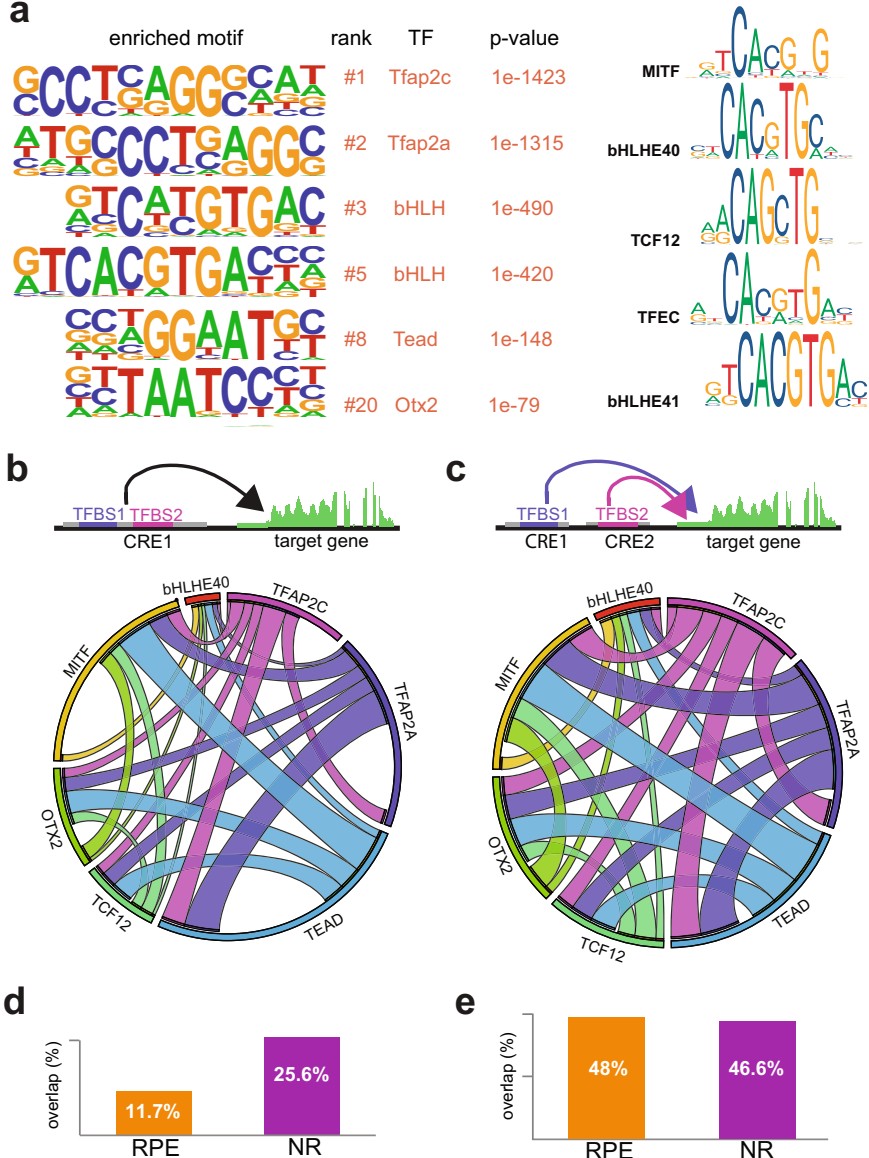

**Fig. 5 RPE motif enrichment analysis. a** Representative TF binding motifs enriched in RPE DOCRs as identified by HOMER. Analysis of the binding motif similarity among TFs of the bHLH family (http://jaspar.genereg.net). **b** Circoplot illustrating the co-occupancy rate of TFBS in the same DOCRs for the main TFs identified in the motif enrichment analysis. **c** Circoplot illustrating the degree of co-regulation between TFs regulating the same gene through different DOCRs. **d** Average percentage of co-occupancy in the same DOCR for two different TF in the RPE and NR. **e** Average percentage of two different TF regulating the same gene through different DOCRs in the RPE and NR.

developmental network. To explore this possibility, we assessed GO enrichment for the genes associated with RPE DOCRs containing the different TFBS and then we grouped the results using hierarchical clustering (Supplementary Fig. 8). The outcome of this clustering approach supported a branched regulatory scenario. Thus, whereas some TF-encoding genes (i.e., *tcf12*, *tfap2c*, *otx2*) appeared associated with many GO terms, others were more specific: such as the case of *mitfa* that was associated only with pigment cell differentiation (Supplementary Fig. 8).

Finally, to investigate the regulatory logic of activating and repressing CREs in each domain, we scanned these regions for enriched motifs separately. Strikingly, those enriched motifs ranking higher for the NR activating CREs, such as the homeobox and sox TF binding sites, were also ranking higher for the repressing regions in this tissue (Fig. 6; Supplementary Data 9). A very similar scenario was observed for the RPE peaks, which

regardless of being activating or repressing regions showed tfap2c, tfap2a, bHLH, and otx2 consensus binding sites as top enriched motifs. When GO terms associated with genes linked to these regions were examined, we observed terms linked to eye morphogenesis enriched for both NR activating and RPE repressing regions (Fig. 6). This finding reveals context-dependent TFs activity and points to a common set of genes activated in the NR and repressed in the RPE by antagonistic GRNs. In fact, a detailed analysis of the gene lists associated with NR activating and RPE repressing regions showed that many of the retinal specifier genes (including *hmx4*, *lhx9*, *mab21l1*, *nr2f2*, *pax6a*, *pax6b*, *rx2*, *six3a*, or *sox2*) are under the antagonistic regulation of the NR and RPE GRNs (Supplementary Data 10). In addition, the analysis of GO term enrichment also suggested domain-specific functions for the NR and RPE GRNs. Thus, NR repressing CREs are associated with genes involved in mesoderm

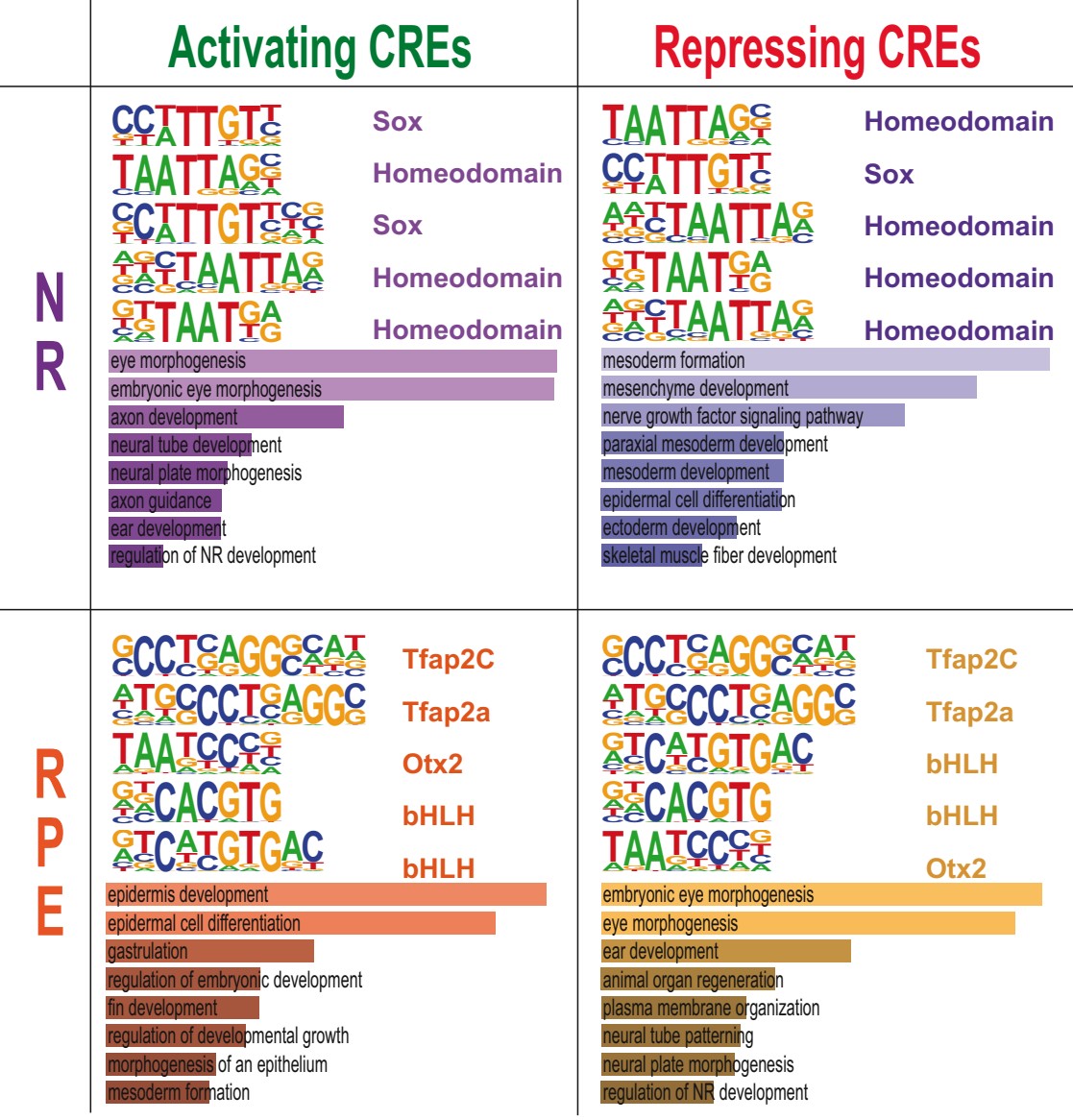

**Fig. 6 Motif enrichment analysis of activating and repressing CREs.** Representative TF motifs enriched in activating (left column) and repressing (right column) CREs in both the NR and RPE domains. Analysis of GO terms enrichment for genes associated with each set of elements is indicated.

formation, whereas RPE activating regions are linked with epidermal differentiation genes.

**Desmosomal components are activated during RPE specification.** The integration of our ATAC-seq and RNA-seq data allows formulating and testing different hypotheses related to genetic programs controlling the specification of retinal tissues. As an example, we followed up the observation that genes encoding for intermediate filaments are enriched in the RPE domain (Supplementary Fig. 7b). Keratin looping into desmosomal plaques plays a fundamental role in maintaining tissue architecture under mechanical load[49] (Fig. 7a). We thus investigated the transcriptional profile of other desmosomal components as the RPE and NR networks diverge. A detailed analysis revealed that not only keratin genes, but also many desmosomal genes such as *dspa*, *evpla*, *pleca*, or *ppl*, are among the most upregulated genes (i.e., highest fold change) in committed RPE cells (Fig. 7b). Many of these genes increased their expression already at 18 hpf, when RPE cells start to differentiate morphologically, but hours before

the tissue acquires pigmentation (Fig. 7b). To gain insight into the regulation of these cytoskeletal components, we performed a motif enrichment analysis of the subset of DOCRs associated with keratins and other desmosomal genes upregulated in the RPE (Supplementary Data 11). This analysis revealed Tead motifs as top-ranked overrepresented binding sites in the CREs regulating desmosomal genes (Fig. 7c). When compared to that of the entire set of DOCRs, the average number of motifs per peak was 4.5 fold higher for those CREs specifically associated with components of the desmosome machinery (1.35 vs. 0.3). This observation suggests a key role for Tead family proteins in the regulation of the intermediate filament cytoskeleton. To test this hypothesis, we took advantage of available zebrafish double mutants for the main Tead coactivators *yap* and *taz*, which have been reported to display RPE differentiation defects[28]. In agreement with our observations, the expression of keratins (*krt4* and *krt8*), as well as pigmentation genes (tyr and tyrp1b), were significantly reduced in *yap* −/− *taz* −/− double mutant head tissue at 18 hpf as determined by RT-qPCR (Fig. 7d). As a control, the expression of

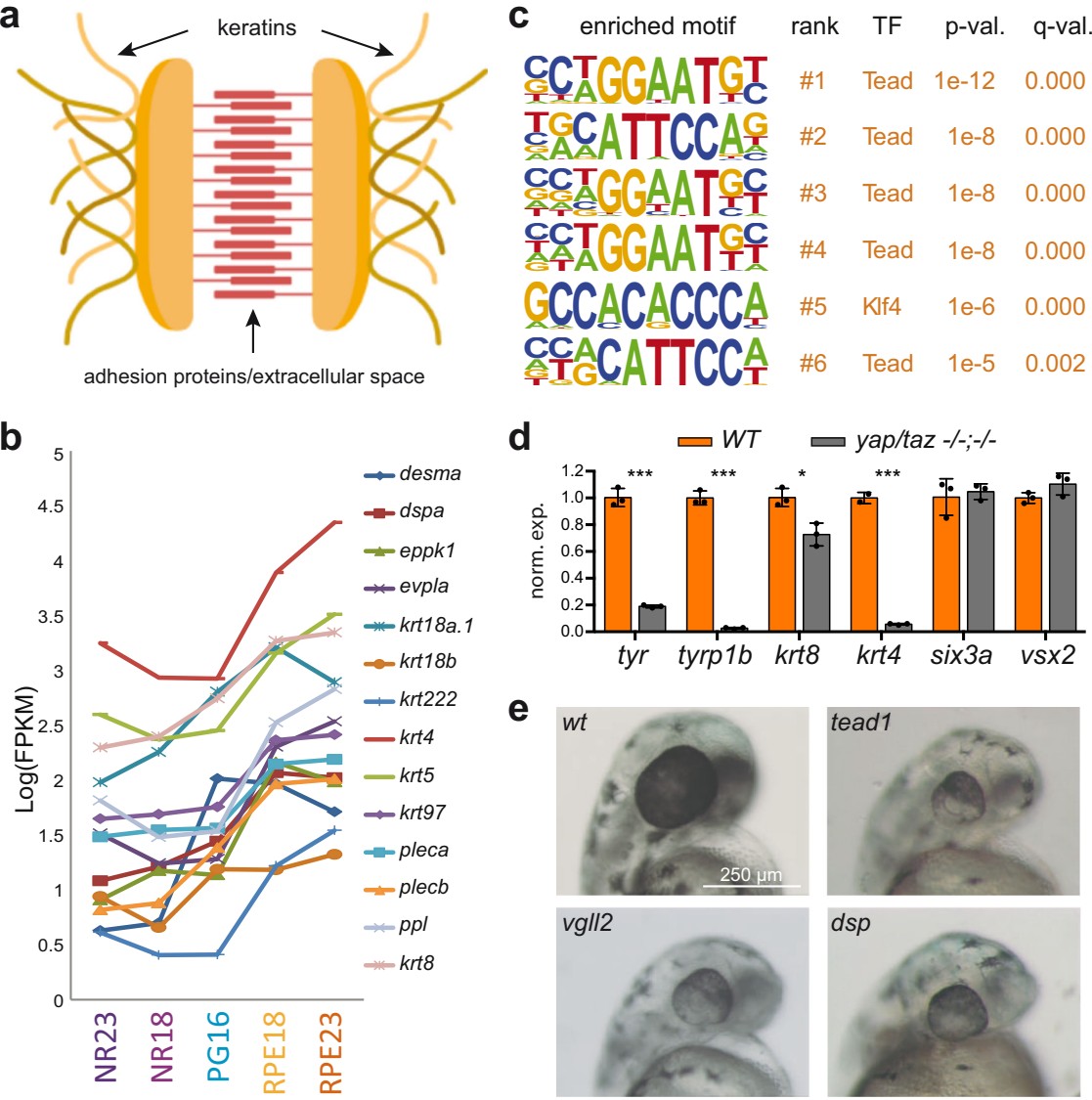

**Fig. 7 Regulation of desmosomal components during RPE specification. a** Schematic representation of a desmosome junction. **b** Intermediate filament and desmosomal gene expression variations during optic cup development. Expression values are reported as log(FPKM). **c** Motif enrichment analysis of the DOCRs associated with genes encoding intermediate filament or desmosome components. **d** mRNA levels of keratin genes as well as RPE and NR markers as determined by RT-qPCR in wild type and *yap −/− taz −/−* double mutant zebrafish samples (dissected heads) at 18 hpf. Significant differences are indicated ($n = 3$; Two-tailed *T*-test; ***=$p < 0.001$; *=$p < 0,05$. p values: *tyr* = 3.4e−5; *tyrp1b* = 6e-6; *krt8* = 1.2e−2; *krt4* = 3.1e−5; *six3a* = 0.67; *vsx2* = 0.12). Data are presented as mean ± SD. Source data are provided as a Source Data file. **e** Representative stereo microscope images of zebrafish embryo heads at 48 hpf: wild type and embryos injected with Cas9 (300 ng/μl) together with the following sgRNAs (80 ng/μl) combinations: *vgll2a* and *vgll2b* (vgll2); *tead1a* and *tead1b* (tead1) and *dspa* and *dspb* (dsp). *Magnification Bar = 250 μm. The same magnification was used for each image in the series.* Note the reduced eye size and RPE hypopigmentation in the crispants. Injections were repeated twice with similar results. See also Supplementary Fig. 9.

the NR markers *vsx2* and *six3a* was unaffected in the mutant tissue. These results further support a role for Tead factors in the transcriptional regulation of desmosomal genes at the RPE.

The high efficiency of the CRISPR/Cas9 technology allows performing F0 mutagenesis screens in zebrafish[50]. We took advantage of this technology to test functionally some of the genes identified in our analyzes. A total of 21 genes, selected on the basis of their expression profile, CRE composition and/or dynamics, associated GO terms, and the number of paralogues, were investigated in an F0 pilot screen. When crispant embryos were examined at 48 hpf, a substantial proportion of the tested genes (i.e., 12 out of 21 sgRNA combinations) displayed eye malformations, such as microphthalmia, eye fissure closure defects, and/or

hypopigmentation (Supplementary Fig. 9; Supplementary Data 12). To further confirm the early role of the candidates as components of the eye GRNs, we examined optic cup architecture and size in the crispants at 24 hpf (Fig. 8 and Supplementary Fig. 10). All candidates displaying eye malformations at 48 hpf also showed a significant reduction of eye size at 24 hpf, with the exception of *nr2f1* (Fig. 8p). None of the injected candidate sgRNAs impaired the general folding of the optic cup, as functional interference with the very early specifier *rx3*[43–45] does (Fig. 8b). This suggests that these candidates do not participate in the initial specification of the optic vesicle. Interestingly, among the sgRNA combinations tested, the injection of those directed against the structural desmosomal components *dspa/b*, the Tead

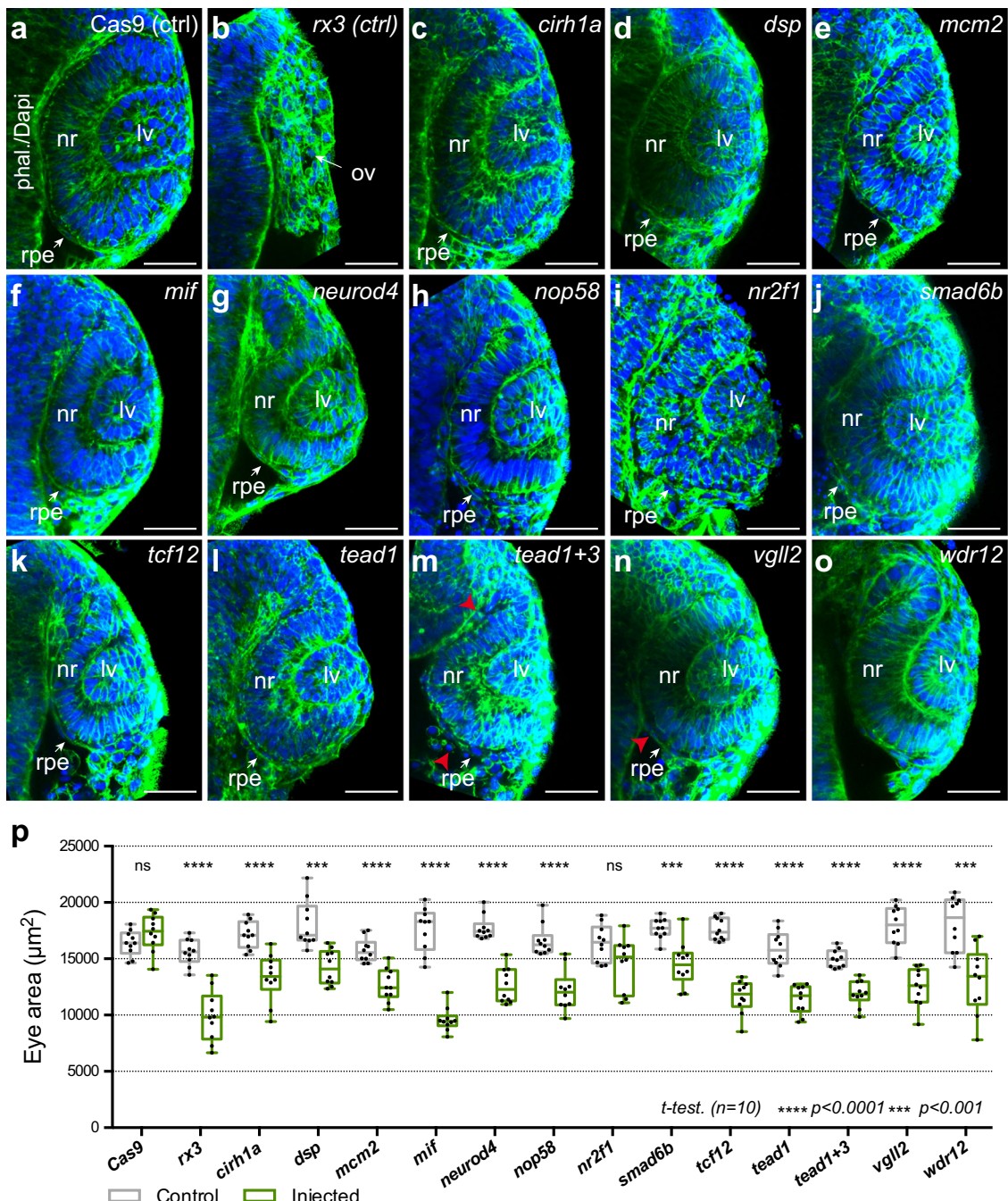

**Fig. 8 Optic cup architecture and size quantification in candidate gene crispant embryos. a–o** Representative optical sections through the optic cup of crispant 24 hpf embryos. Phalloidin/DAPI staining reveals tissue organization of control embryos injected only with Cas9 (**a**), with sgRNAs directed against *rx3* (**b**), or with sgRNAs for the different candidate genes (indicated in (**c**)–(**o**)). Note that, in contrast to *rx3* crispant embryos, the optic cup is formed in all cases. However, quantitative analysis of eye area (*p*) reveals a significant reduction of optic cup size for all candidates tested with the exception of *nr2f1*. Individual values (*n* = 10) are plotted in front of standard box-and-whiskers (Two-tailed *T*-test); ***=*p* < 0.001, ****=*p* < 0.0001. *p* values: *cas9* = 0.161; *rx3* = 1.15e−6; *crh1a* = 8.67e−5; *dsp* = 1.82e−4; *mcm2* = 3.09e−5; *mif* = 1.05e-9; *neurod4* = 9.00e−8; *nop58* = 8.09e−6; *nr2f1* = 0.053; *smad6b* = 2.25e−4; *tcf12* = 2.71e−9; *tead1* = 1.62e−6; *tead1* + *3* = 1.15e−6; *vgll2* = 9.15e−7; *wdr12* = 5.00e−4. Abnormal morphology of the RPE cells (m,n) is indicated (red arrowheads). ov optic vesicle, lv lens vesicle. Bar = 50 µm. Source data are provided as a Source Data file. See also Supplementary Figs. 9 and 10.

regulators *vgll2a/b*, and particularly the cocktail *tead1a/b* + *tead3a/b* resulted in a significant proportion of the F0 embryos displaying hypopigmentation (Fig. 7, and Supplementary Fig. 9), reduced eye size, and abnormal RPE morphology (Fig. 8, and Supplementary Fig. 10). These results further indicate that both desmosomal assembly and Tead activity are required for RPE and optic cup morphogenesis.

**Gene expression analysis during hiPSCs-to-RPE differentiation.** A second important finding derived from our gene expression clustering analysis was the identification of two waves of transcriptional regulators during the specification of the RPE in zebrafish. Understanding this TF recruitment sequence in humans may have important basic and translational applications. We thus asked whether the same *consecutio temporum* identified

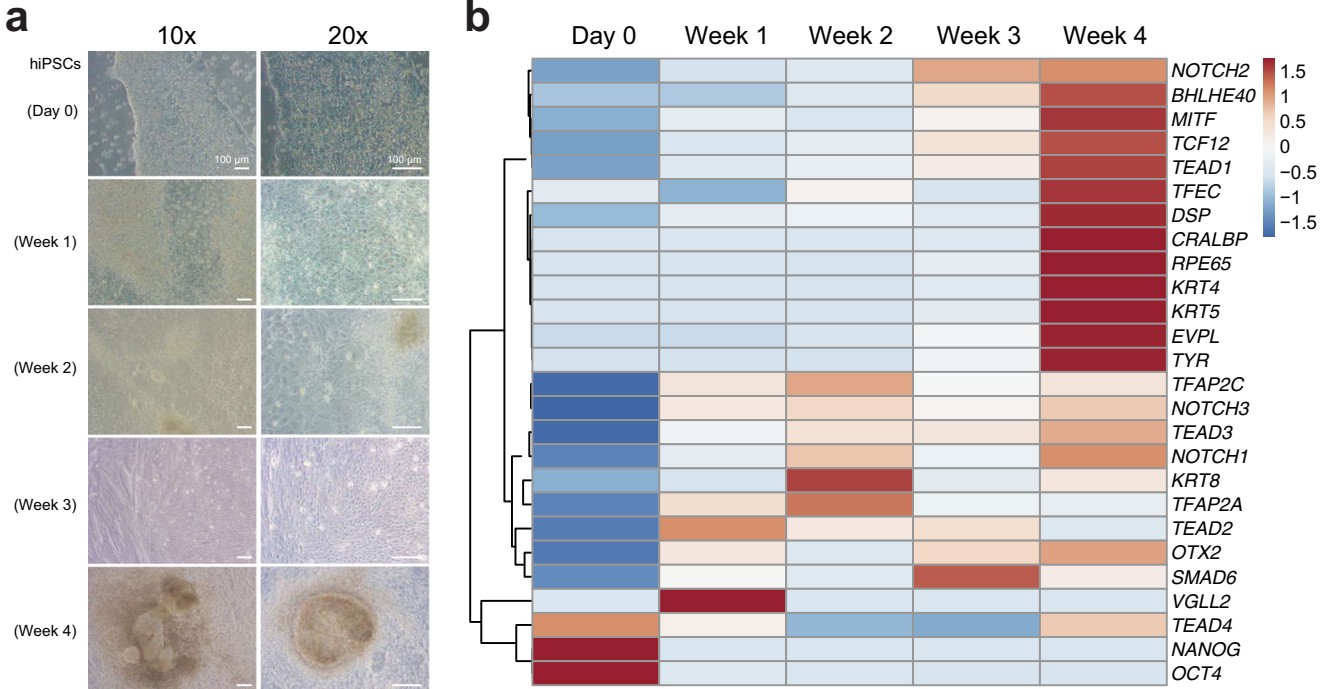

**Fig. 9 Gene expression during hiPSCs-to-RPE differentiation. a** Bright field microscopy images (10× and 20×) of hiPSCs before (Day 0) and during their differentiation towards RPE. Note the progressive acquisition of epithelial morphology and pigmentation (observed in three independent experiments). **b** Hierarchical clustered heatmap showing gene expression level variations, as determined by RT-qPCR during the differentiation towards RPE from hiPSCs. Note the conservation of the two clusters (early and late transcriptional waves) identified in zebrafish.

in zebrafish was conserved in human iPSCs differentiating to RPE. During the first four weeks of differentiation in culture from hiPSCs (see "Methods"), pluripotent stem cells progressively change their morphology to a cobblestone appearance, acquiring a light pigmentation at the end of the fourth week (Fig. 9a). Using RT-qPCR, we tested mRNA levels for a total of 26 RPE genes including genes encoding for stemness markers (*NANOG* and *OCT4*); mature RPE markers (i.e., *CRALBP, RPE65*, and *TYR*); known RPE specifiers activated in the second wave of gene expression (i.e., *BHLHE40, MITF, OTX2*, and *TFEC*); desmosomal components (i.e., *DSP, EVPL, KRT4, KRT5, KRT8*); and TFs and signaling molecules activated in the first wave of gene expression (*NOTCH1, NOTCH2, NOTCH3, SMAD6, TCF12, TEAD1, TEAD2, TEAD3, TEAD4, TFAP2A, TFAP2C*, and *VGLL2*) (Fig. 9b; Supplementary Data 13). This analysis showed that most of the human genes orthologous to the zebrafish early specifiers cluster together according to their expression profiles, and reach maximal expression within the first weeks of culture. This was the case for *SMAD6, TEAD1, TEAD3, TFAP2A, TFAP2C*, and particularly for *VGLL2 and TEAD2*, the expression of which peaked very transiently. Similarly, the human orthologs of most of the genes in zebrafish were identified as activated in a second transcriptional wave (including *MITF, BHLHE40, TFEC, CRALBP, RPE65*, and *TYR*) also clustered together reaching a maximum expression level in the fourth week of culture. These findings indicated that despite the very different time scales of the developmental programs in these far-related vertebrate species, hours in zebrafish, and days in humans, their RPE regulatory networks share a common logic of two waves in the TFs recruitment sequence.

## Discussion
Here we have used a double RNA-seq/ATAC-seq approach to zoom into the bifurcation of the NR/RPE developmental programs, which will give rise to specialized neurons and pigmented squamous cells respectively. The transient nature of this differentiation process, as well as the limiting size of the cell populations involved[6], have hindered any systemic approach to investigate the architecture of the specification networks as they branch. Using tissue-specific transgenic lines, we have isolated NR and RPE precursors by FACS. Although expression driven by the RPE-specific element *E1_bHLHe40* has also been reported in the ciliary marginal zone and periocular neural crest cells, this is only reported after optic cup folding[38]. Therefore, only minor contamination, if any, would be expected in RPE precursors isolated at 23 hpf. The combination of isolated cell populations with the high sensitivity and deep sequencing coverage of the RNA-seq and ATAC-seq methods[51], allowed us to overcome previous constraints: permitting not only the detection of transcriptomic variations and active cis-regulatory modules but also helping to define hierarchical relationships among the core components of the network.

In our analyses, we have used a proximity method to associate genes and OCRs. Previous studies have confirmed that this link to the adjacent gene is correct in 90% of the cases[52]. Therefore, although this estimate should be taken cautiously when individual CREs are examined, it is a valid approach when conclusions are driven from genome-wide analysis. In this study, we identified approximately 30,000 chromatin regions differentially open (DOCRs) between the NR and RPE domains. The examination of their methylation status, as well as their overlap with H3K27ac and H3K4me3 marks, indicates that most of these regions (≈95%) correspond to initially inactive enhancers that get progressively demethylated and activated at the phylotypic period, a behavior reported for developmental genes in general[39]. A much smaller cluster (≈5%) matches hypomethylated constitutively active promoter regions (Supplementary Figs. 2 and 4). It is therefore likely that most of the regions here identified as DOCRs correspond to active CREs. A specific example of a distal enhancer is provided for the *nr2f2* locus (Supplementary Fig. 11).

Our data, together with previous observations by others, are consistent with the hypothesis that the NR is the default state of the optic vesicle precursors. Morphologically, optic vesicle precursors either at the medial or the lateral layers share a similar neuroepithelial character: i.e., elongated cells polarized along the apicobasal axis arranged in a pseudo-stratified epithelium[6,7]. During optic cup morphogenesis, NR precursors retain this neuroepithelial morphology, whereas differentiating RPE cells undergo profound cell shape changes as they progressively flatten into a squamous epithelium[12]. These observations correlate in our analyzes with a larger number of differentially up-regulated genes and differentially opened chromatin regions at the RPE (Supplementary Fig. 2). We have shown that repressing CREs associated with TFs dominate in the RPE network, whereas activating elements are more abundant in the NR program (Fig. 2). Furthermore, the motifs analysis of activating elements in the NR and repressing elements in the RPE indicates that they are antagonistically regulating a similar set of genes. It is therefore very likely that the acquisition of the RPE fate requires the global repression of the NR program. However, although the role of *Vsx2* in the suppression of RPE identity via direct *Mitf* repression has been well-documented[5,18,20], the mechanisms underlining the repressive activity of the RPE network are much less understood[53]. Interestingly, *Pax6* seems to cooperate both with NR and RPE specifiers acting as a balancing factor between the two networks depending on the cellular context[54,55]. Our motif analysis also points to the same set of TFs acting as repressors or activators within the same tissue, suggesting the chromatin environment conditions their activity. This is in line with previous work showing that TFs have diverse regulatory functions depending on the enhancer context[56].

One of our main observations is that NR and RPE networks are remarkably robust. This is the case particularly in teleosts, in which whole-genome duplication resulted in additional copies of eye specifiers. In zebrafish, several homeobox TFs expressed in the NR converge on the 5′–TAATT–3′ motif, including vsx1, vsx2, rx1, rx2, rx3, lhx2b, lhx9, hmx1, and hmx4. Our motif discovery predictions suggest these proteins, together with sox factors, may co-regulate the same genes and cooperate within the same CREs, although these are theoretical predictions that may require validation by direct ChIP-seq studies. Supporting a co-regulation, the mutation of genes encoding for retinal homeobox TFs, such as *rx2* in medaka[57] or *lhx2* in zebrafish[58] does not compromise severely the identity of the tissue. In the RPE, redundancy pertains mainly to the core bHLH motif (5′–CACGTG–3′), which is shared by mitfa, bHLHE40, bHLHE41, and tfec. In agreement with this, the simultaneous elimination of *mitfa* and *mitfb* has no apparent consequences on RPE specification or pigmentation in zebrafish[24]. All these observations further point to a broad redundancy and cooperativity among the main nodes of the eye specification networks.

As expected, cell shape changes associated with the acquisition of the NR and RPE identities have a clear reflection at the transcriptomic level. The differentiation of the NR entails the activation of cytoskeletal components associated with microtubules polarization, in agreement with the neuroepithelial character of the tissue[7]. In contrast, here we show that a significant number of keratins and other desmosomal genes get recruited in the presumptive RPE hours before this tissue acquires its distinctive pigmentation. Although keratins and plaque proteins have been reported as RPE markers in several vertebrate species[59], their contribution to the morphogenetic program of this tissue remains unexplored. Given the role of desmosomal plaques in conferring resistance to mechanical load[49], it is tempting to hypothesize that the activation of these cytoskeletal genes may be linked to RPE precursors adapting their cytoskeleton to increased tissue tension

as they flatten. Motif enrichment analysis of CREs linked to desmosomal genes, together with the genetic evidence here provided by *yap* −/−; *taz* −/− double mutants and Tead crispants, support a role for Tead TFs in the activation of keratin genes at the RPE. Indeed, several keratin (e.g., *krt5*, *krt8*, and *krt97*), and desmosomal genes (e.g., *evpla*, *pleca*, *plecb*, and *ppl*) have been identified as targets for Yap/Tead complexes by ChIP-seq studies in mammalian cells[60] and DamID-seq studies in zebrafish embryos[61]. Thus, the direct transcriptional regulation of keratins and plaque genes by Yap/Tead complexes seems a conserved theme across tissues and vertebrate species.

Arguably one of the most notable observations in this study is the relatively late peak of expression of genes such as *mitfa* or *otx2*, which were previously considered among the earliest RPE specifiers[47]. Their delayed peak of expression occurs after the specification and flattening of RPE precursors have commenced, and rather coincides with the onset of pigmentation in the tissue at 23 hpf. This agrees with previous studies indicating a cooperative role for Otx and Mitf in the direct activation of the melanogenic gene battery[23]. A detailed expression study in zebrafish also reported two separated phases for RPE specification, the first of which is *mitf* independent[36]. Here we show that the early phase of specification entails the recruitment of a different set of transcriptional regulators, including *smad6b*, *tead1b*, *tead3b*, *tfap2a*, *tfap2c*, *tcf12*, and *vgll2b*. Some of these TFs, or their cofactors, act as upstream regulators of Otx and Mitf, such is the case for *Tcf*[62] and Tead-cofactors *Yap* and *Taz*[28,29]. Others, such as *tfap2a*, play an important role in RPE specification, although their hierarchical role within the RPE GRN remains unclear[48]. Finally, here we also describe a similar sequence of TF recruitment in human differentiating RPE cells. Thus, despite the different morphology of the cells (i.e., cuboidal in humans and squamous in zebrafish) and the large evolutionary distance (≈450 mya), the regulatory logic that specifies the RPE seems largely conserved across vertebrates.

Our results may be relevant to identify causative genes for eye hereditary diseases, as mutations in many nodes of the eye GRNs result in congenital eye malformations[63]. Importantly, our findings uncover an unanticipated regulatory logic within the RPE specification network. This provides critical information to improve hiPSCs-to-RPE differentiation protocols, a key step in cell replacement strategies for retinal degenerative diseases.

## Methods

**Fish maintenance**. The zebrafish (*Danio rerio*) AB/Tübingen (AB/TU) wild-type strains, the transgenic lines *tg(vsx2.2:GFP-caax)*[8] and *tg(E1_bHLHe40:GFP)*[38] (Supplementary Fig. 12) and the mutant strains *yap^mw48* and *taz^mw49* were maintained as heterozygous stocks[28]. Animal experiments were carried out according to ethical regulations. Experimental protocols have been approved by the Animal Experimentation Ethics Committees at the Pablo de Olavide University and CSIC (license number 02/04/2018/041).

**Cell cytometry**. Zebrafish cells were dissociated and prepared for FACS as previously described[64]. PG at 16 hpf, NR at 18, and 23 hpf was isolated from dissected heads of the *tg(vsx2.2:caax-GFP)*. RPE at 18 and 23 hpf were isolated from whole *tg (E1_bHLHE40:GFP)* embryos. A FACSAriaTM Fusion flow cytometer (BD Facs-Diva Software 8.0.1) was used to recollect only the GFP + cells (Supplementary Data 14). GFP + cells were isolated directly in Trizol for RNA extraction, or in ATAC-seq tagmentation buffer for open chromatin detection.

**RNA extraction**. Total RNA was extracted using 750 ul TRIzol LS (Invitrogen) following the manufacturer's protocol. Possible DNA contamination was eliminated by treating the RNA samples with TURBO DNAse-free (Ambion). The concentration of the RNA samples was evaluated by Qubit (Thermo Fisher), and then the samples were used for subsequent applications.

**qPCR**. cDNA retrotranscription and qPCR were performed as described[61]. For primer sequences, see Supplementary Tables 1 and 2. *HPRT1* and *GAPDH* were

used as housekeeping genes for human samples, whereas *eef1a1l1* was used for zebrafish samples.

**RNA-seq**. RNA was extracted from sorted cells and then treated with DNAse as described above. rRNAs were eliminated from the samples with Ribo-Zero® rRNA Removal Kit (Illumina) prior to library preparation. Samples were sequenced in SEx125bps or PEx125bps reads with an Illumina Hiseq 2500. We obtained at least 35 M reads from the sequencing of each library. Three biological replicates were used for each analyzed condition (Supplementary Data 15 for correlations between replicates). Reads were aligned to the danRer10 zebrafish genome assembly using Tophat v2.1.0. Transcript abundance was estimated with Cufflinks v2.2.1. Differential gene expression analysis was performed using Cuffdiff v2.2.1, setting a corrected $p$-value < 0.05 as the cutoff for statistical significance of the differential expression. Multidimensional scaling analysis (MDS) was performed using the function MDSplot of the CummeRbund package in R 3.6.1. Soft clustering of time-series gene expression data was done for all the transcripts with variance among the five conditions ≥3 using the R package Mfuzz with $m = 1.5$[65]. The TF transcript subset was extracted from the total list of genes using the tool "Classification System" of PANTHER[66] filtering for the protein class "transcription factors" (PC00218). Some TFs not present in the database for annotation issues (i.e., mitfa, vsx1, vsx2, rx1, rx2, rx3, lhx2b, hmx1, hmx4, sox21a) were added manually. The cytoskeleton component subset was obtained retrieving all the genes belonging to GO term "cytoskeleton" (GO:0005856), including all the child and further descendant GO terms, with biomaRt. Hierarchical clustering was defined with R function hclust with agglomeration method "complete". Heatmap representation was obtained using the R package pheatmap. Transcriptomic variations were plotted using exclusively transcripts that resulted to be differentially expressed from the comparison between at least two of our experimental conditions. TFs and cytoskeleton components were filtered using the same methodology for Mfuzz clustering. Gene ontology analysis was performed with the online tool GOrilla[67] or Panther[66] using two unranked lists of genes (target and background lists).

**ATAC-seq**. ATAC-seq was performed starting from 5000 sorted cells using a FAST-ATAC protocol previously described[68]. All the libraries were sequenced 2 × 50 bp with an Illumina Hiseq 2500 platform. We obtained at least 100 M reads from the sequencing of each library. For data comparison, we used two biological replicates for each condition. Reads were aligned to the danRer10 zebrafish genome using Bowtie2[69] with -X 2000—no-mixed—no-unal parameters. PCR artifacts and duplicates were removed with the tool rmdup, available in the Samtools toolkit[70]. In order to detect the exact position where the transposase binds to the DNA, read start sites were offset by $+4/-5$ bp in the plus and minus strands. Read pairs that have an insert < 130 bp were selected as nucleosome-free reads. Differential chromatin accessibility was calculated as reported[71]. All chromatin regions reporting differential accessibility with an adjusted $p$-value < 0.05 were considered as DOCRs. The average size of these regions was 422.54 and 499.56 nucleotides for the RPE and NR respectively. All the DOCRS have been associated with genes using the online tool GREAT[72] with the option "basal plus extension". Gene ontology analysis of all the genes associated with DOCRs was also performed with GREAT. De novo motif enrichment of TF binding sequences in the sets of DOCRs was performed using HOMER[73]. Top enriched TF PWMs from the HOMER results and PWMs from the JASPAR database[74] were used as input for the online tool FIMO[75] to assess the exact TFBS genome position in the DOCRs. Before estimating the rate of TF co-occupancy in the same peak among the binding motifs for the different TFs, all the binding motif sequences overlapping for more than 3 bps were eliminated, keeping only the TF binding sequences with the lowest $p$-value.

**Activating/repressing CRE configuration**. For this analysis, only the DEGs and the DORCs that could be associated with each other were taken into account. The log2FoldChange values of transcript expression and chromatin accessibility of NR and RPE at 23 hpf were used to discriminate four clusters of activating and repressing CREs. Gene Ontology enrichment analysis of the genes belonging to the different clusters was performed with FishEnrichr[76].

**CRE 5mC analysis**. Whole-genome bisulfite sequencing and TET-assisted bisulfite sequencing[39] data were trimmed with Trimmomatic software[77] and mapped onto the danRer10 genome assembly using WALT[78] keeping only the reads mapping to a unique genomic location. Duplicates were removed with sambamba[79] and DNA methylation levels were calculated using MethylDackel (https://github.com/dpryan79/MethylDackel). All heatmaps were made using Deeptools[80] and Gene ontology enrichments were calculated with GREAT[72]. ATAC-seq reads were counted using BEDtools[81] and statistics were performed in R.

**CRISPR/Cas9 F0 screening**. All the sgRNAs were designed using the online tool CRISPscan (https://www.crisprscan.org/)[82] and synthesized following described protocols[83]. All the sgRNAs were selected to target the first half of the CDS in exons resulting actually expressed in the eye tissues from our RNA-seq data (trying to avoid the first exon to prevent the usage of an alternative start codon that would produce a possibly functional protein), with an efficiency score > 58 and no predicted off-targets (Supplementary Table 3). Two different sgRNAs were used

together to target the same gene. The sgRNAs were injected in the zebrafish yolk at the 1-cell stage at a final concentration of 80 ng/μl together with the Cas9 endonuclease at a concentration of 300 ng/μl. In total, 1 nl of the mixture was injected into each embryo. In the case a target gene had a close paralogue, the sgRNAs targeting both of the paralogues were injected at the same time, adjusting the final concentration of the sgRNA-Cas9 mixture. The mutagenic efficiency of the sgRNAs was assessed in individual DNA samples from CRISPR-Cas9 micro-injected embryos at 24 hpf (Supplementary Data 16). Targeted genomic regions were amplified using specific primers designed to generate 200 to 600 base pairs amplicons. When no difference was observed between wild-type and microinjected amplicons, a T7-endonuclease I assay was performed. For this, the PCR products were purified (NucleoSpin Plasmid QuickPure, Macherey-Nagel) and re-annealed (95 °C, 5 min; 95 °C to 85 °C at −2 °C/s. and 85 °C to 25 °C at −0.1 °C/s.). Then, the samples were digested using T7 endonuclease-I (NEB) at 37 °C for 1 h and 30 min and loaded in a 2% agarose gel for analysis. Lethality, phenotypic features, and penetrance were assessed at 24 and 48 hpf. Retinal histology was revealed with nuclear DAPI (1:5000, Sigma) and Phalloidin-Alexa488 (1:20 Invitrogen) staining incubated in PBT overnight at 4 °C. Selected embryos were mounted in 35 mm FluoroDish plates (WPI, FD3510-100) using 1% low melting agarose and examined in confocal microscopy (Zeiss LSM 880) using 25× or 40× multi-immersion objectives. All pictures were processed using FIJI/ImageJ (Version 1.50i).

**Cell culture**. Human hiPSCs were obtained, after informed consent, from peripheral blood monocytes by cell reprogramming using a non-integrative Sendai virus vector as described[84]. The study was granted ethical approval number PR-01-2015 from the Andalusian Ethics Committee for Research with Embryonic and iPS Cells (Comité Andaluz de Ética de Investigación con muestras biológicas de naturaleza embrionaria y otras células semejantes). Before sample collection, a clinician and a researcher provided detailed information to the donor about the procedure, the research project, the implications for his/her health, and his /her rights on the biological sample and its derivatives. Then, the donor, the clinician, and the researcher signed an Informed Consent sheet. The primary biological sample was obtained at the University Hospital Virgen Macarena (Seville, Spain) with the participation of the Public Health System Biobank (BBSPA), which generated the codes for the pseudonymization of the samples. The donor provided 4 mL of peripheral blood using a regular arm venipuncture procedure. From the fresh blood sample, mononuclear cells were enriched and reprogrammed. Cells were maintained in feeder-free adherent conditions onto Matrigel-covered plates in standard incubation at 37 °C, 5% $CO_2$, 20% $O_2$. hiPSCs were fed every two days with mTser1 serum-free culture medium and passaged every 5-7 days depending on confluency. Dispase was used for gentle dissociation for passage. A 6-well plate with undifferentiated, well-grown hiPSCs was the starting point of the experiment (Day 0). To induce RPE differentiation culture medium was changed to the following: KO DMEM, KSR 15%, Glutamax 2 mM, non-essential aminoacids 0.1 mM, β-mercaptoethanol 0.23 mM, Penicillin/streptomycin. Differentiating cells were harvested directly in Trizol LS (Invitrogen) at day 0, and weeks 1, 2, 3, and 4 for gene expression studies.

**Reporting summary**. Further information on research design is available in the Nature Research Reporting Summary linked to this article.

## Data availability
Datasets are available in the Gene Expression Omnibus (GEO) repository under the following accession numbers: RNA-seq "GSE150346" and ATAC-seq "GSE150189". PWMs were retrieved from the JASPAR database (http://jaspar.genereg.net/). All other relevant data supporting the key findings of this study are available within the article and its Supplementary Information files or from the corresponding author upon reasonable request. Source data are provided with this paper. A reporting summary for this article is available as a Supplementary Information file. Source data are provided with this paper.

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

## Acknowledgements

We thank Lázaro Centanin, Juan Tena, Miguel Moreno-Mateos, Joaquin Letelier, Marta Magri, and Santiago Negueruela for their excellent scientific advice and critical input; Corin Díaz and Katherina García for their technical assistance with FACS; and Ana Férnández-Miñan and Laura Romero for their help in transgenesis assays. This work is supported by the following grants: (I) To J.-R.M.-M.: From the Spanish Ministry of Science, Innovation, and Universities (MICINN): BFU2017-86339P with FEDER funds, MDM-2016-0687 and PY20_00006/Junta de Andalucía. (II) To O.B. Australian Research Council (ARC) Discovery Project (DP190103852). (III) To F.-J.D.-C.: Andalusian Ministry of Health, Equality and Social Policies (PI-0099-2018). (IV) To P.B.: BFU2016-75412-R with FEDER funds; PCIN-2015-176-C02-01/ERA-Net Neuron ImprovVision, and a CBMSO Institutional grant from the Fundación Ramón Areces. (V) To both J.-R. M.-M. and P.B.: BFU2016-81887-REDT, as well as Fundación Ramón Areces-2016 (Supporting L.B.).

## Author contributions

L.B. conducted most experiments, performed bioinformatic analyzes, and had the main contribution in figures and manuscript editions. J.C., E.S.-R., and M.A.-C. contributed to the CRISPR-Cas9 F0 screen and the phenotypic analysis of the crispants. S.N., T.M.-M., and R.P. contributed to protocols refining, FACS experiments, and fish maintenance. B.d.C. and F.-J.D.-C. carried out hiPSCs experiments together with L.B. O.B. performed 5mC analysis and contributed to the manuscript edition. J.-R.M.-M. conceived the project and assisted L.B. in data analysis. The manuscript was edited and written by J.-R.M.-M. and P.B., who co-supervised all the work.

## Competing interests

The authors declare no competing interests.
