## [Peer Review File · Nature Communications]

REVIEWER COMMENTS

Reviewer #1 (Remarks to the Author):

The manuscript by Buono et al. describes RNA-seq and ATAC-seq datasets generated in neural retina (NR) and retinal-pigmented epithelium (RPE) cells isolated from the zebrafish eye in order to investigate the gene networks governing the NR/RPE specification in the vertebrate eye. They describe TF binding motifs in differentially open chromatin regions and identify differentially expressed genes that could be involved in NR/RPE bifurcation.

At present, the manuscript is descriptive. It does not provide new insights into the molecular mechanisms of RPE/NR specification but only a description of datasets. In my opinion, follow up validation experiments demonstrating new mechanisms are needed for publication in Nature Communications.

Major concerns:

-There is no information about the purity of isolated RPE and NR cells and insufficient data proving that the lines used for isolating RPE and NR populations (E1_bHLHe40:GFP and vsx2.2:GFP) at 18hpf and 23hpf are properly restricted to RPE and NR. Moreover the reference given for the E1_bHLHe40:GFP line in the method section (Moreno-Marmol et al., 2020) does not exist. The authors should show confocal sections for both markers at 18hpf (in addition to 23hpf). Furthermore they could show in situ hybridization of endogenous markers of RPE and NR to compare with their confocal sections and therefore demonstrate their restricted patterns. Finally they should show in their RNA-seq data that the bhlhe40/vsx2 genes and other known markers are strongly different between RPE and NR.

-We have no information about the reproducibility of RNA-seq replicates. This should be shown in the form of pairwise correlations or hierarchical clustering of replicates.

-The authors test the hypothesis that Tead transcription factors regulate keratin and other desmosomal genes in RPE using yap/taz mutant lines. Given that yap/taz mutant lines lose RPE cells, it is expected that all genes enriched in RPE are downregulated in the heads of these lines. Furthermore it is reported in Miesfeld et al., 2015 that yap^{-/-}; taz^{-/-} embryos arrest before RPE development. Therefore the downregulation of genes could simply be a consequence of failed RPE differentiation but does not necessarily demonstrate a direct role of Tead TFs in the transcriptional regulation of genes. ChIP experiments or knockdown of Tead TFs in RPE cells are needed.

-In Miesfeld et al., 2015, several yap mutants were used but the authors do not say which one they used in this study. In general, information about zebrafish lines is missing in the method section.

-The motif analysis on DOCRs and clustering of gene expression is interesting. It confirms known TFs and reveals potential novel TFs or genes (for example desmosomal components) involved in RPE/NR specification. However, without further validation experiments it remains only descriptive. One would expect detailed validation in follow up experiments for at least one candidate using imaging and zebrafish mutant lines.

-The CRISPR/Cas9 screen (Fig S9) is very preliminary. What criteria were used to select these 21 genes? There is no validation of the efficiency of gRNAs nor any detailed description of the potential eye phenotypes. Again, detailed analyses on mutant lines would be needed to make solid conclusions on one or more candidates. Usually, the control for CRISPR is not a wild-type fish but a control with

either only the Cas9 or a control sgRNA with the Cas9. Authors could have used these controls instead of taking a wild-type fish.

-The human data in hIPSCs-to-RPE does not bring much to the paper. It would have been more interesting to perform RNA-seq/ATAC-seq in iPSCs-to-RPE differentiation and compare to iPSCs-to-retina differentiation to determine if gene networks are conserved.

Reviewer #2 (Remarks to the Author):

This study by Lorena Buono et al., focuses on global characterization of the transcriptional and epigenomic changes that occur during the dynamic process of early cell-fate specification in the developing eye. They focus on early stages when a common pool of neural progenitor cells bifurcate and give rise to two important lineages of the eye: the neuroretinal (NR) progenitors and the precursors of the retinal pigmented epithelium (RPE). To accomplish their goal, the authors elegantly combined RNA-seq and ATAC-seq on sorted populations of cells isolated from zebrafish embryos. The integration of the transcription profile with open chromatin sites resulted in an overview of the regulatory regions and predicted regulated genes in different stages and lineages.

This is a discovery-based approach that provides a comprehensive view of the regulatory network in each lineage over time. One of the interesting finding from the study is enrichment of the TEAD binding motif among the keratins and the increase in other desmosomal genes in the RPE compared to the NR. This suggests an important role for TEAD proteins in regulating important cytoskeletal and adhesion genes in the RPE.

The main concern is that most of the identified and discussed factors are known players in the early specification of these eye lineages, including Yap/Taz, as indicated by the authors. This supports the quality of the data but provides only limited new insight for our understanding of the events that control the bifurcation of common neural progenitors to the distinct eye lineages. A knockdown screen for some of the genes is mentioned, but the analysis of the phenotypes is not comprehensive and thus it is difficult to conclude on the significance of the findings. Adding analyses of a critical regulator or enhancer region (possibly one of the CRE sites that contains the TEAD motif), identified and predicted by their unbiased approach, would be an important addition for the validation of the many interesting predictions from this study.

There are several issues that need further clarification:

Comments:

1. The authors used two transgenic lines to isolate the cells by FACS based on GFP expression. The lines were Vsx2:GFP, which is active early on in all of the neural progenitors of the eye and is later restricted to the NR, and bHLHe40:GFP, which is expressed in the RPE progenitors. Three stages were sampled with Vsx2:GFP—16, 18 and 23 hpf, and two stages (18 and 23) with bHLHe40:GFP. The GFP-expressing cells from these transgenes formed the basis for the subsequent analyses. Thus, it is very important to present the activity of these transgenes at the stages that were sampled and include the whole embryo. Fig. 1 presents the eye with these transgenes only at 23 hpf. The differences in expression between stages could reflect sampling of additional regions of the embryo.

2. In the results, explain the criterion for assigning CRE to a putative target gene; is it the distance from the TSS, or the nearest gene that is active in one of the stages/tissues?

3. The analysis suggests to the authors (page 7) that transcription factors (TFs) undergo more complex regulation than all other genes based on the number of CREs associated with the genes. The

statistical significance supporting this conclusion is not clear. TFs belong to one defined functional group, whereas the "other genes" are heterogeneous, including those encoding enzymes, adhesion proteins and non-coding RNAs; it is therefore not clear to me what the appropriate control should be for determining complexity in gene regulation, and how this can be supported by a statistical analysis. Possibly performing this analysis on cytoskeletal genes (or another functional category) would provide a better comparison.

4. The authors state (page 7) that more TFs are activated in the NR than in the RPE, where TF expression is globally repressed. This may merely reflect a difference in the complexity of the two tissues. The NR at this early stage is a heterogeneous population of progenitors destined to multiple retinal fates, whereas the RPE precursors are destined to a single fate. It is possible that more TFs are active in the NR because the bulk RNA-seq of the NR represents TFs from several different lineages. This needs to be clarified/discussed.

5. The authors conclude that RPE fate needs the repression of early expressed TFs that remain active in the NR. The evidence that the repressed TFs in the RPE are those that remain active in the NR is not clear. Are these TFs expressed in the OV and are maintained in NR and inhibited in the RPE? or these TFs are expressed in one of the lineages but not in the bipotential progenitors of the OV.

6. The clustering analyses in Fig. 3 should include information on the similarity between genes in a cluster (homogeneity score) and the differences between clusters and number of genes in each cluster. It is important to clarify the criteria for assigning the genes to different clusters.

7. It is stated that "the rate of co-occupied peaks dropped from 25.6% in the NR to 11.7% in the RPE (Fig. 5D)": Clarify how the individual peaks were defined. What is the average size of the peaks in each tissue? Could it be that the differences are due to the different landscapes—in RPE there are less open regions? Maybe the average length of CRE is reduced? What is the statistical test to determine whether this is a significant difference?

Additional comments:

Page 7: "...and this was also accompanied by a higher average fold change (OF TRANSCRIPTS?) in the RPE than in the NR associated DOCRs." This sentence needs to be clarified.

Fig. 7 shows a motif analysis of the desmosomal genes. How many genes were used for this analysis? Indicate the P-values (these values should be presented as in Fig. 5).

Reviewer #3 (Remarks to the Author):

I have reviewed the manuscript by Buono et al which reports a multi-omics analysis of early eye development in zebrafish, examining the bifurcation of the retina and RPE lineages during early developmental stages. These tissue types arise from common progenitors and while some excellent embryological and developmental studies have been performed over several decades, the regulatory logic of how this split happens remains unclear. Thus, the work fits a nice niche in the field. Overall, I quite liked the manuscript. While I have a few nits to pick about some of the data and presentation, I think the manuscript makes a nice contribution.

While there is a lot to like about the manuscript and most certainly, the datasets will be useful for others to do complementary analyses, I think the most noteworthy aspect of the study to me is the strong support for retina as a base state of these common progenitors and that a retina program must

be repressed for RPE to form. Indeed, this fits quite nicely with a lot of published work but the genomics support here is quite strong and from several angles.

I do see several things that should be addressed by the authors, some less significant than others, and presented here in no particular order.

1) I found the first paragraph of the introduction related to evolution and association between pigment and photoreception to be quite forced. This isn't needed here and is distracting at the start; the data, or at least how they are analyzed, don't really address this issue, so I'd strongly suggest just omitting this and start with a strong developmental focus.

2) I'd like some more information/data about the specificity of the bhlhe40 transgenic used for FACS isolations. Does this only label RPE cells or do periorbital mesenchyme or neural crest also express it? If so, this has some bearing on the results.

3) I found the logic presentation of the methylation and hydroxymethylation section difficult to follow. While these have been studied to some extent in the eye, particularly hydroxymethylation, and also in zebrafish (see Seritrukul and Gross, 2017), I couldn't easily follow what the authors were trying to do here. I think this section of text may need a bit of work.

4) My most serious concern is that I am unconvinced by the F0 CRISPR screen results. While this technique is exciting, perhaps because it is a practical way to screen candidates, I am dubious here that the assay is as clean as it needs to be. For nearly every target for which a phenotype is shown, the embryos look very sick - all are microphthalmic, have pigmentation defects and other anterior neural defects. Many appear to be edemic and have cell death throughout their bodies and many also have colobomas. The percentages of dead embryos are also quite high. These phenotypes can also be ascribed to a lack of specificity. I am sure that some of these are bona fide targets, but I don't think the data support this hypothesis. I understand that most, if not all, of these genes are also expressed non-ocularly, but it is difficult to believe that loss of each in this paradigm leads to the same suite of phenotypic defects. More is needed to be sure that the conclusions are supported.

5) I also have some problems with the hiPSC data. While I am not an expert here, these data seemed forced to me, in two ways. The first is in an attempt to try to drive human relevance of the study. This is nice, of course, when it works but here it seems out of place. The second is more in the interpretation itself that the late "phase" of peak expression and that this is functionally separate from the early specification phase. While this could be true, it could also be that the weekly sampling frequency missed high expression at another time or that the level of some of these factors might not need to be that high to drive these early events but to trigger and/or maintain the differentiation program, expression needs to be higher. Put differently, I don't see why one assumes that the peak expression disallows any important function in a non-peak period.

Reviewer #1 (Remarks to the Author):

The manuscript by Buono et al. describes RNA-seq and ATAC-seq datasets generated in neural retina (NR) and retinal-pigmented epithelium (RPE) cells isolated from the zebrafish eye in order to investigate the gene networks governing the NR/RPE specification in the vertebrate eye. They describe TF binding motifs in differentially open chromatin regions and identify differentially expressed genes that could be involved in NR/RPE bifurcation. At present, the manuscript is descriptive. It does not provide new insights into the molecular mechanisms of RPE/NR specification but only a description of datasets. In my opinion, follow up validation experiments demonstrating new mechanisms are needed for publication in Nature Communications.

R1: We thank this reviewer for the critical comments. The focus of our work was the systemic characterization of the GRNs involved in the NR/RPE bifurcation. We think that a detailed functional characterization of each of the candidate genes and regulatory regions identified in this work is beyond the scope of our study. However, we strengthen validation experiments by improving the F0 CRISPR screen analysis in the revised version (see specific comments below).

Major concerns:

-There is no information about the purity of isolated RPE and NR cells and insufficient data proving that the lines used for isolating RPE and NR populations (E1_bHLHe40:GFP and vsx2.2:GFP) at 18hpf and 23hpf are properly restricted to RPE and NR. Moreover the reference given for the E1_bHLHe40:GFP line in the method section (Moreno-Marmol et al., 2020) does not exist. The authors should show confocal sections for both markers at 18hpf (in addition to 23hpf). Furthermore they could show in situ hybridization of endogenous markers of RPE and NR to compare with their confocal sections and therefore demonstrate their restricted patterns.

R2: We have previously characterized the two transgenic lines used in the study and they are specific of each domain at 18hpf and 23hpf. The vsx2.2:GFP-caax line has been extensively described in previous studies from the group (Bogdanovic et al 2012, Dev Cell; Gago-Rodrigues et al 2015 Nat Comm; and Nicolás-Pérez et al 2016, eLife). Particularly, time-lapse analysis of this line through the developmental window considered shows that it is specific of the NR domain (Nicolás-Pérez et al 2016, eLife). The bhlhe40 line used is specific of the RPE precursors at the early stages considered in our study. It is only later in development, after the complete folding of the OC at 24 hpf, that GFP expression is detected also in the ciliary marginal zone (CMZ), the pineal gland and few neural crest cells surrounding the eye. This is clearly shown by time-lapse analysis in our previous study Moreno-Marmol et al. 2020, BioRxiv (doi.org/10.1101/2020.09.23.310631). Although a reference to this paper was included in the main text, the full reference of the work was not annotated in the final references list. We apologize for this mistake that has been corrected in the new version of the manuscript.

To make this clearer, we show the dynamic expression of the transgenes in Figure S12. We think including this data as a main figure will be redundant with previous publications, as dynamic data (movies) have been provided in Nicolas-Pérez et al 2016, eLife and Moreno-Marmol et al 2020, BioRxiv (currently under final review in eLife). In addition, we are now including datasets showing the isolation of the RPE and NR populations by FACS (Supplementary Dataset S14).

Finally they should show in their RNA-seq data that the bhlhe40/vsx2 genes and other known markers are strongly different between RPE and NR.

R3: We are showing this information in the manuscript. Differential gene expression is clearly shown in Figure 3B: see fpkm levels in clusters 1 (including vsx2 and other NR markers) and 4 (including bhlhe40 and other RPE markers).

-We have no information about the reproducibility of RNA-seq replicates. This should be shown in the form of pairwise correlations or hierarchical clustering of replicates.

R4: In the revised version we have included the information showing the correlation between the RNA-seq replicates (Supplementary Dataset S15). These correlations were always high (i.e. average = 0.96 ± 0.03).

-The authors test the hypothesis that Tead transcription factors regulate keratin and other desmosomal genes in RPE using *yap/taz* mutant lines. Given that *yap/taz* mutant lines lose RPE cells, it is expected that all genes enriched in RPE are downregulated in the heads of these lines. Furthermore it is reported in Miesfeld et al., 2015 that *yap*^{-/-}; *taz*^{-/-} embryos arrest before RPE development. Therefore the downregulation of genes could simply be a consequence of failed RPE differentiation but does not necessarily demonstrate a direct role of Tead TFs in the transcriptional regulation of genes. CHIP experiments or knockdown of Tead TFs in RPE cells are needed.

R5: In the manuscript we have examined the expression of keratins and other RPE markers at 18 hpf in double mutants *yap*^{-/-}; *taz*^{-/-}. Despite development is arrested from 21-22 hpf on in double mutant embryos, the optic vesicle is formed at 18hpf (Miesfeld et al 2015 and our own observations). In agreement neural retina markers (*six3* and *vsx2*) are expressed at levels similar to that of wild type embryos at that stage (Figure 7D). Furthermore, according to our observations RPE precursors are present at 18 hpf in *yap*^{-/-}; *taz*^{-/-} double mutant embryos. This was assessed by ISH against *tfec*, which appeared reduced but not absent in double mutants (see figure for the referees R1).

We agree with the referee in that, without further support, may be difficult to interpret the qPCR experiments in the mutants. The fact that Yap/Tea complexes are early specifiers of the RPE (together with keratin/desmosomal genes being among the earliest genes activated) raises the question to which extent are they directly activated by Yap/Tea or indirectly through other early RPE specifiers. However, two independent evidences suggest that it could be a direct regulation. The first is the enrichment, here reported, for Tead motifs in the regulatory regions associated to genes encoding for desmosomal components. The second is the evidence for a direct binding of *yap/tead* complexes to these genomic regions, as determined by CHIP-seq in mammalian cells (Estarras et al, 2017, Genes Dev; Lian et al., 2010, Genes Dev; and Zanconato et al., 2015, Nat Cell Biol), and by Dam-ID seq in zebrafish embryos (our own data in Vazquez-Marin et al., 2019, Development). Taking all together, we think that our claim “These results further support a role for Tead factors in the transcriptional regulation of desmosomal genes at the RPE” is a valid statement.

-In Miesfeld et al., 2015, several *yap* mutants were used but the authors do not say which one they used in this study. In general, information about zebrafish lines is missing in the method section.

R5: We apologize for overlooking this. The alleles used in the study are *yap1*^{mw48} and *taz*^{mw49}. This information is now available in the methods. As we have also included the missing reference for the bHLHe40:GFP line

(Moreno-Marmol et al. 2020, BioRxiv), the four zebrafish lines used in the study are now referenced to previous work.

-The motif analysis on DOCRs and clustering of gene expression is interesting. It confirms known TFs and reveals potential novel TFs or genes (for example desmosomal components) involved in RPE/NR specification. However, without further validation experiments it remains only descriptive. One would expect detailed validation in follow up experiments for at least one candidate using imaging and zebrafish mutant lines.

-The CRISPR/Cas9 screen (Fig S9) is very preliminary. What criteria were used to select these 21 genes? There is no validation of the efficiency of gRNAs nor any detailed description of the potential eye phenotypes. Again, detailed analyses on mutant lines would be needed to make solid conclusions on one or more candidates. Usually, the control for CRISPR is not a wild-type fish but a control with either only the Cas9 or a control sgRNA with the Cas9. Authors could have used these controls instead of taking a wild-type fish.

R6 (a similar answer was included in the response letter to referee #3): We have now strengthen this analysis. Nevertheless, we would like to make the following considerations regarding the analysis itself, its relative weight in our manuscript and the new experiments provided.

1- As pointed by referee#3, F0 CRISPR screens provide a practical way to perform a preliminary screen of the candidates. In the manuscript this was intended as a pilot study on the functionality of selected genes. The results were included as a supplementary figure (S9), as we considered that they were not essential to sustain any of the main claims of the work. That being said, we do agree with the reviewers in that we could have done a better job. In the revised version we have improved this analysis by validating the mutagenic efficiency of the sgRNAs, quantifying the resulting microphthalmia phenotypes, and adding a histological analysis of the crispant retinæ at 24 hpf. These results are now included in two additional figures (Figs. 8 and S10) and a complementary dataset for the validation of the mutagenic efficiency of the sgRNAs (Dataset S16).

2- Regarding the doubts raised on the specificity of the sgRNAs: It is important to consider that the genes selected for the screen (21) were predicted as excellent candidates: i.e. according to their expression profile, CRE composition and/or dynamics, associated GO terms, and number of paralogs. Despite this, 43% of the sgRNA combinations injected did not result in an obvious ocular phenotype. Thus, general toxicity associated to the sgRNAs or Cas9 can be ruled out, and these injections can be used as a negative control.

3- To further improve the specificity of our previous observations at 48 hpf (Fig. S9), we have repeated the sgRNA injections and examined the embryos also at 24hpf (Fig. S10), thus reducing the accumulative impact of indirect effects (i.e. cell death) on the development of the optic cup. We have examined retinal histology (by phalloidin/DAPI staining) and measured eye size in the crispants (Figs. 8 and S10). As indicated by the referee, we are also including both “positive controls” (i.e. embryos injected with an sgRNA directed against *rx3*) and a negative control (i.e. embryos injected only with Cas9). The new quantitative analyses confirmed our previous observation of microphthalmia as a common feature in the affected retinas (Figure 8). This is not that surprising as human mutations in well-known retinal specifiers including *RX*, *PAX6*, *VAX2*, *VSX2*, *OTX2*, and *MITF*, (also known as the Coloboma Gene Network; Gregory-Evans et al. 2013) often lead to microphthalmia, regardless if they are expressed in the NR or RPE domain.

-The human data in hiPSCs-to-RPE does not bring much to the paper. It would have been more interesting to perform RNA-seq/ATAC-seq in iPSCs-to-RPE differentiation and compare to iPSCs-to-retina differentiation to determine if gene networks are conserved.

R7: We kindly disagree with the referee on the relevance of our observations on hiPSCs. We think that the expression data in human cells validates some of our results in a different vertebrate species, and may be useful to improve hiPSCs-to-RPE differentiation protocols. Of course a full RNA-seq/ATAC-seq analysis on hiPSCs will add a nice comparative study in the future, but this is definitely beyond the scope of our work. We believe that a similar expression profile clustering of key TFs in different species (as we show in Figures 3 and 8) strongly argues for a conserved cis-regulatory logic. For these reasons, we believe that the hiPSCs data should be maintained in the paper.

Reviewer #2 (Remarks to the Author):

This study by Lorena Buono et al., focuses on global characterization of the transcriptional and epigenomic changes that occur during the dynamic process of early cell-fate specification in the developing eye. They focus on early stages when a common pool of neural progenitor cells bifurcate and give rise to two important lineages of the eye: the neuroretinal (NR) progenitors and the precursors of the retinal pigmented epithelium (RPE). To accomplish their goal, the authors elegantly combined RNA-seq and ATAC-seq on sorted populations of cells isolated from zebrafish embryos. The integration of the transcription profile with open chromatin sites resulted in an overview of the regulatory regions and predicted regulated genes in different stages and lineages. This is a discovery-based approach that provides a comprehensive view of the regulatory network in each lineage over time. One of the interesting finding from the study is enrichment of the TEAD binding motif among the keratins and the increase in other desmosomal genes in the RPE compared to the NR. This suggests an important role for TEAD proteins in regulating important cytoskeletal and adhesion genes in the RPE.

The main concern is that most of the identified and discussed factors are known players in the early specification of these eye lineages, including Yap/Taz, as indicated by the authors. This supports the quality of the data but provides only limited new insight for our understanding of the events that control the bifurcation of common neural progenitors to the distinct eye lineages. A knockdown screen for some of the genes is mentioned, but the analysis of the phenotypes is not comprehensive and thus it is difficult to conclude on the significance of the findings. Adding analyses of a critical regulator or enhancer region (possibly one of the CRE sites that contains the TEAD motif), identified and predicted by their unbiased approach, would be an important addition for the validation of the many interesting predictions from this study.

R1: We thank this reviewer for the positive comments on our work. Here we provide a systemic approach to investigate globally the architecture of the NR and RPE specification networks as they branch. Our main conclusions come from this global analysis of the networks: e.g. findings on networks' redundancy, on the NR program as a "default" state, or on the key recruitment of desmosomal genes in the RPE. In this sense, we do agree that the main focus of the work has not been the identification of new players in the NR and RPE specification. Although we have identified/investigated many "novel genes", including a few transcriptional regulators with no previous link to NR or RPE specification (e.g. *neurod4*, *nr2f1*, or *vgll2*), our work suggests that most of the main upstream nodes of the NR and RPE GRNs may have been already identified by genetic studies. We think this is a very interesting observation by itself.

-Regarding functional validation assays (see also the reply to other referees), we have done a substantial effort to improve our CRISPR-Cas9 F0 screen. In the original manuscript this was intended as a pilot study on the functionality of selected genes. The results were included as a supplementary figure (S9), as we considered that they were not essential to sustain any of the main claims of the work. That being said, we do agree with the reviewers in that we could have done a better job. In the revised version we have improved substantially the analysis by validating the mutagenic efficiency of the sgRNAs, quantifying the resulting microphthalmia phenotypes, and adding a histological analysis of the crispant retinae at 24 hpf. These results are now included in two additional figures (Figs. 8 and S10) and a complementary dataset (Dataset S16).

- Regarding the characterization of cis-regulatory regions, we do agree with this referee that it would have added to the manuscript. Following the referee suggestion, we tested a few (8) of these regions in transgenesis assays using the zebrafish enhancer detection (ZED) vector (Bessa et al 2009). Particularly, a region 90 Kb away from *nr2f2* 3' showed expression specifically at the optic cup in F0. We are now including the description of this distal enhancer in Figure S11. Unfortunately, due to problems in our fishroom (i.e. related both to limitations imposed by Covid19 and unexpected larval lethality), we have not succeeded in obtaining the corresponding stable lines. Nevertheless, we could perform the converse test: examining chromatin accessibility for known NR and RPE specific elements. Interestingly, the endogenous regions corresponding to the *bhlhe40* enhancer and the *vsx2* promoter used to generate the lines *tg(E1_bHLHe40:GFP)* and *tg(vsx2.2:caaxGFP)* are associated to DOCRs specific of each domain (Figure S12). Interestingly, *bHLHe40* has been proved to be a direct yap/taz target in several Chip-seq and Dam-ID studies (Vazquez-Marin et al), and its E1 enhancer (i.e. peak 24355; chr11:35535287-35536336) contains the TEAD motif (322-3331; p value

7.05e-5). We think that taken together these observations provide sufficient validation for our ATAC-seq results.

There are several issues that need further clarification:

Comments:

1. The authors used two transgenic lines to isolate the cells by FACS based on GFP expression. The lines were *Vsx2:GFP*, which is active early on in all of the neural progenitors of the eye and is later restricted to the NR, and *bHLHe40:GFP*, which is expressed in the RPE progenitors. Three stages were sampled with *Vsx2:GFP*—16, 18 and 23 hpf, and two stages (18 and 23) with *bHLHe40:GFP*. The GFP-expressing cells from these transgenes formed the basis for the subsequent analyses. Thus, it is very important to present the activity of these transgenes at the stages that were sampled and include the whole embryo. Fig. 1 presents the eye with these transgenes only at 23 hpf. The differences in expression between stages could reflect sampling of additional regions of the embryo.

R2: We have previously characterized the two transgenic lines used in the study and they are specific of each domain at 18hpf and 23hpf. The *vsx2.2:GFP-caax* line has been extensively described in previous works from the lab as a NR specific line (Bogdanovic et al 2012, Dev Cell; Gago-Rodrigues et al 2015 Nat Comm; and Nicolás-Pérez et al 2016, eLife). Particularly, time-lapse analysis of this line through the developmental window considered shows that it is specific of the NR domain (Nicolás-Pérez et al 2016, eLife). The *bhlhe40* line used is specific of the RPE precursors at the early stages considered in our study. It is only later in development, after the complete folding of the OC at 24 hpf, that GFP expression is detected also in the ciliary marginal zone (CMZ), the pineal gland and few neural crest cells surrounding the eye. This is clearly shown by time-lapse analysis in our previous paper Moreno-Marmol et al. 2020, BioRxiv (doi.org/10.1101/2020.09.23.310631). Although a reference to this paper was included in the main text, the full reference of the work was not annotated in the final references list. We apologize for this mistake that has been corrected in the new version of the manuscript.

To make this clearer, we show the dynamic expression of the transgenes in Figure S12. We think including this data as a main figure will be redundant with previous publications, as dynamic data (movies) have been provided in Nicolas-Pérez et al 2016, eLife and Moreno-Marmol et al 2020, BioRxiv.

2. In the results, explain the criterion for assigning CRE to a putative target gene; is it the distance from the TSS, or the nearest gene that is active in one of the stages/tissues?

R3: This information was already included in the methods section: “All the DOCRS have been associated with genes using the online tool GREAT with the option basal plus extension”. According to this association rule each gene’s regulatory domain is then extended up to the nearest upstream and downstream genes, but no longer than 1 Mb in each direction (McLean et al. 2010; Nat Biotechnology). As we mentioned in the discussion, this is a proximity method that predicts correct associations in 90% of the cases (Yoshida et al, 2019, Cell). For the assignment of DOCRs all genes were taken into account, not only the up-regulated/down-regulated genes. However, as shown in Fig 1D, 47% of the associated genes correspond to differentially regulated genes at 23 hpf.

3. The analysis suggests to the authors (page 7) that transcription factors (TFs) undergo more complex regulation than all other genes based on the number of CREs associated with the genes. The statistical significance supporting this conclusion is not clear. TFs belong to one defined functional group, whereas the "other genes" are heterogeneous, including those encoding enzymes, adhesion proteins and non-coding RNAs; it is therefore not clear to me what the appropriate control should be for determining complexity in gene regulation, and how this can be supported by a statistical analysis. Possibly performing this analysis on cytoskeletal genes (or another functional category) would provide a better comparison.

R4: In the revised version we have assessed the statistical significance of our analysis using a Chi-square test with Yates correction. Comparing directly TFs vs cytoskeletal genes would have been produced a statistical bias due to the different number of genes involved (359 TFs; 134 cytoskeletal genes). To overcome this issue, we

compared both TFs and cytoskeletal genes versus a randomized control containing the same exact number of genes and extracted from the total list of DEGs associated with DOCRs. In our experiment, the greater association of CRE/gene for TF vs. a randomized control resulted to be significant with p-value < 0.0001. When the same analysis was performed for cytoskeletal genes, the test returned a not significant p-value ($p = 0.3347$). This information has been included in the text.

4. The authors state (page 7) that more TFs are activated in the NR than in the RPE, where TF expression is globally repressed. This may merely reflect a difference in the complexity of the two tissues. The NR at this early stage is a heterogeneous population of progenitors destined to multiple retinal fates, whereas the RPE precursors are destined to a single fate. It is possible that more TFs are active in the NR because the bulk RNA-seq of the NR represents TFs from several different lineages. This needs to be clarified/discussed.

R5: At the stages considered in our study neurogenesis has not started in the NR domain (as stated in page 5). In fact, we were very careful to avoid differentiation stages, as this would have represented a further branching of the NR program into the different neuronal lineages. At the early stages considered (i.e. before 24hpf), retinal precursors are still uncommitted and remain multipotent, as it has been reported by lineage tracing studies in several vertebrates (Turner & Cepko, 1987, Nature; Holt et al, 1998, Neuron; Wetts and Fraser, 1998, Science; Turner et al, 1990, Neuron), including zebrafish (He et al, 2012, Neuron). Therefore, it is extremely unlikely that a differential activation of TFs in the NR vs the REP reflects an early restriction of neuronal lineages.

In addition, our data on the activation/repression of TFs in the NR/RPE (Figure 2) argues for a different regulatory logic between the two tissues, but not necessarily for a more complex architecture in the NR gene regulatory network. Complexity can be achieved either through activating or repressing regions, and a similar number of regulatory regions are associated to TFs in each tissue (1562 in the NR vs 1575 in the RPE). Although this is in our opinion a very interesting discussion, it has not been included in discussion due to space limitations.

5. The authors conclude that RPE fate needs the repression of early expressed TFs that remain active in the NR. The evidence that the repressed TFs in the RPE are those that remain active in the NR is not clear. Are these TFs expressed in the OV and are maintained in NR and inhibited in the RPE ? or these TFs are expressed in one of the lineages but not in the bipotential progenitors of the OV.

R6: This evidence is provided in the manuscript with specific examples: "In fact, a detailed analysis of the gene lists associated to NR activating and RPE repressing regions showed that many of the retinal specifiers themselves (including *hmx4*, *lhx9*, *mab21l1*, *nr2f2*, *pax6a*, *pax6b*, *rx2*, *six3a*, or *sox2*) are under the antagonistic regulation of the NR and RPE GRNs (Dataset S10)". Many of these NR TFs are already expressed in the bipotential progenitors, though a lower levels. Thus, as the networks bifurcate, they get activated in the NR and repressed in the RPE. To illustrate this antagonism, fpkm values are shown in Figure 3 and Dataset S1.

6. The clustering analyses in Fig. 3 should include information on the similarity between genes in a cluster (homogeneity score) and the differences between clusters and number of genes in each cluster. It is important to clarify the criteria for assigning the genes to different clusters.

R7: To our knowledge homogeneity scores apply only to k-means clustering but not to hierarchical clustering, which seeks to build a hierarchy without having a fixed number of clusters. Hierarchical clustering was defined with R function `hclust` with agglomeration method "complete". Heatmap representation was obtained using the R package `pheatmap` (This information has been incorporated to methods). The number of genes in each cluster is now indicated in figures 3. Their identity was already provided in Dataset S5, where we are now also including an expanded dendrogram.

7. It is stated that "the rate of co-occupied peaks dropped from 25.6% in the NR to 11.7% in the RPE (Fig. 5D)": Clarify how the individual peaks were defined. What is the average size of the peaks in each tissue? Could it be that the differences are due to the different landscapes—in RPE there are less open regions? Maybe the

average length of CRE is reduced? What is the statistical test to determine whether this is a significant difference?

R8: - In the methods section we refer to a previously published approach for peak calling and differential chromatin accessibility (Magri et al 2020, Front. Cell Dev. Biol.). Within this pipeline, peak calling is achieved using the standard tool MACS2 (Zhang et al. 2008, Genome Biol).

- The average size of the peaks is 422.54 and 499.56 nucleotides for the RPE and NR respectively. This information is now included in methods.

- The large differences observed in the rate of co-occupancy cannot be explained by a reduced number of open regions in the RPE. In fact, "we observed a larger number of DOCRs in the RPE than in the NR (18909 vs 11263; adjusted p-value <0.05)", as indicated in the text.

- Finally, although the difference in average size (422.54 vs 499.56) is statistically significant (Paired t-test, p-value < 2.2e-16), it is not that clear to which extent this 16% larger size could affect by two-fold the co-occupancy rate. In fact, it could very well be the other way around, a higher co-occupancy rate impinging on chromatin accessibility. Either way, we believe this would have little impact on in the general conclusions of the paper (i.e. pervasive redundancy both in the RPE and NR networks), and due to space limitations we decided not to include it in the discussion.

Additional comments:

Page 7: "...and this was also accompanied by a higher average fold change (OF TRANSCRIPTS?) in the RPE than in the NR associated DOCRs." This sentence needs to be clarified.

R9: We have change the sentence to the following:

"Regardless the adjusted p-value, we observed a larger number of DOCRs in the RPE than in the NR (18909 vs 11263; adjusted p-value <0.05), and their average chromatin accessibility fold change was higher than that of NR associated DOCRs (Fig. S2B,C)."

Fig. 7 shows a motif analysis of the desmosomal genes. How many genes were used for this analysis? Indicate the P-values (these values should be presented as in Fig. 5).

R10: A total of 108 DOCRs were used in this analysis. These DOCRs resulted to be associated with 11 upregulated desmosomal genes (11 out of a total of 22 desmosomal genes up-regulated in RPE, reflecting the roughly 50% association between DOCRs and DEGs that we estimated in Fig 1 D). This is clearly indicated in Dataset S11, in which both q-values and p-values are shown. Despite the small sample size p-values are highly significant (ranging between 1e-12 to 1e-5) for the top ranked overrepresented binding sites. We are now including p and q values in Fig 7.

Reviewer #3 (Remarks to the Author):

I have reviewed the manuscript by Buono et al which reports a multi-omics analysis of early eye development in zebrafish, examining the bifurcation of the retina and RPE lineages during early developmental stages. These tissue types arise from common progenitors and while some excellent embryological and developmental studies have been performed over several decades, the regulatory logic of how this split happens remains unclear. Thus, the work fits a nice niche in the field. Overall, I quite liked the manuscript. While I have a few nits to pick about some of the data and presentation, I think the manuscript makes a nice contribution.

While there is a lot to like about the manuscript and most certainly, the datasets will be useful for others to do complementary analyses, I think the most noteworthy aspect of the study to me is the strong support for retina as a base state of these common progenitors and that a retina program must be repressed for RPE to form. Indeed, this fits quite nicely with a lot of published work but the genomics support here is quite strong and from several angles.

I do see several things that should be addressed by the authors, some less significant than others, and presented here in no particular order.

R1: We thank this reviewer for the critical comments and suggestions, and appreciate the positive words on the relevance of our study. We have made an effort to resolve all conflicting points. In particular, we have strengthened/clarify the FO CRISPR screen analysis/interpretation as indicated by the referee.

1) I found the first paragraph of the introduction related to evolution and association between pigment and photoreception to be quite forced. This isn't needed here and is distracting at the start; the data, or at least how they are analyzed, don't really address this issue, so I'd strongly suggest just omitting this and start with a strong developmental focus.

R2: We have shortened the introduction to eliminate the unnecessary information. This has helped in focusing the introduction and maintaining the word limits in the text.

2) I'd like some more information/data about the specificity of the bhlhe40 transgenic used for FACS isolations. Does this only label RPE cells or do periocular mesenchyme or neural crest also express it? If so, this has some bearing on the results.

R3: The bhlhe40 line used is specific of the RPE precursors at the early stages considered in our study. It is only later in development, after the complete folding of the OC at 24 hpf, that GFP expression is detected also in the ciliary marginal zone (CMZ), the pineal gland and few neural crest cells surrounding the eye. This is clearly shown in our paper Moreno-Marmol et al. 2020, BioRxiv (doi.org/10.1101/2020.09.23.310631). Although a reference to this paper was included in the main text, the full reference of the work was not annotated in the references list. We apologize for this mistake that has been corrected in the new version of the manuscript.

To make this clearer, we show the dynamic expression of the transgenes in Figure S12. We think including this data as a main figure will be redundant with previous publications, as dynamic data (movies) have been provided in Nicolas-Pérez et al 2016, eLife and Moreno-Marmol et al 2020, BioRxiv.

3) I found the logic presentation of the methylation and hydroxymethylation section difficult to follow. While these have been studied to some extent in the eye, particularly hydroxymethylation, and also in zebrafish (see Seritrakul and Gross, 2017), I couldn't easily follow what the authors were trying to do here. I think this section of text may need a bit of work.

R4: We agree with the referee in that the logic of this section was difficult to follow. We have changed the text to remove unnecessary information and emphasize the main message of the section: i.e. that methylome profiles are useful to show that the differentially open chromatin regions correspond to active cis-regulatory elements (CREs).

Methylome profiles allowed clustering the NR and RPE DORCS in to two groups: (i) a large group (95%) corresponding to active enhancers (K27ac), which get progressively demethylated as development proceeds; and (ii) a small group (5%) of constitutive hypo-methylated promoters (K4me3). We think that this adds

important information on the features of the identified open chromatin regions (DOCRs).

4) My most serious concern is that I am unconvinced by the F0 CRISPR screen results. While this technique is exciting, perhaps because it is a practical way to screen candidates, I am dubious here that the assay is as clean as it needs to be. For nearly every target for which a phenotype is shown, the embryos look very sick - all are microphthalmic, have pigmentation defects and other anterior neural defects. Many appear to be edemic and have cell death throughout their bodies and many also have colobomas. The percentages of dead embryos are also quite high. These phenotypes can also be ascribed to a lack of specificity. I am sure that some of these are bona fide targets, but I don't think the data support this hypothesis. I understand that most, if not all, of these genes are also expressed non-ocularly, but it is difficult to believe that loss of each in this paradigm leads to the same suite of phenotypic defects. More is needed to be sure that the conclusions are supported.

R5: We have now strengthened this analysis that was also considered as preliminary by referee #1. Several considerations are relevant regarding the analysis itself, its relative weight into the manuscript, and the new experiments provided.

1- As pointed by referee#3, F0 CRISPR screens provide a practical way to perform a preliminary screen of the candidates. In the manuscript this was intended as a pilot study on the functionality of selected genes. The results were included as a supplementary figure (S9), as we considered that they were not essential to sustain any of the main claims of the work. That being said, we do agree with the reviewers in that we could have done a better job. In the revised version we have improved this analysis by validating the mutagenic efficiency of the sgRNAs, quantifying the resulting microphthalmia phenotypes, and adding a histological analysis of the crispant retinæ at 24 hpf. These results are now included in two additional figures (Figs. 8 and S10) and a complementary dataset for the validation of the mutagenic efficiency of the sgRNAs (Dataset S16).

2- Regarding the doubts raised on the specificity of the sgRNAs: It is important to consider that the genes selected for the screen (21) were predicted as excellent candidates: i.e. according to their expression profile, CRE composition and/or dynamics, associated GO terms, and number of paralogs. Despite this, 43% of the sgRNA combinations injected did not result in an obvious ocular phenotype. Thus, general toxicity associated to the sgRNAs or Cas9 can be ruled out. In addition, as pointed by the referee, the expression of these genes is not restricted to the retina, and therefore broader phenotypes are expected upon interference.

3- To further improve the specificity of our previous observations at 48 hpf (Fig. S9), we have repeated the sgRNA injections and examined the embryos also at 24hpf (Fig. S10), thus reducing the accumulative impact of indirect effects (i.e. cell death) on the development of the optic cup. We have examined retinal histology (by phalloidin/DAPI staining) and measured eye size in the crispants (Figs. 8 and S10). As indicated by the referee, we are also including both "positive controls" (i.e. embryos injected with an sgRNA directed against *rx3*) and a negative control (i.e. embryos injected only with Cas9). The new quantitative analyses confirmed our previous observation of microphthalmia as a common feature in the affected retinas (Figure 8). This is not that surprising as human mutations in well-known retinal specifiers including *RX*, *PAX6*, *VAX2*, *VSX2*, *OTX2*, and *MITF*, (also known as the Coloboma Gene Network; Gregory-Evans et al. 2013) often lead to microphthalmia, regardless if they are expressed in the NR or RPE domain.

5) I also have some problems with the hiPSC data. While I am not an expert here, these data seemed forced to me, in two ways. The first is in an attempt to try to drive human relevance of the study. This is nice, of course, when it works but here it seems out of place. The second is more in the interpretation itself that the late "phase" of peak expression and that this is functionally separate from the early specification phase. While this could be true, it could also be that the weekly sampling frequency missed high expression at another time or that the level of some of these factors might not need to be that high to drive these early events but to trigger and/or maintain the differentiation program, expression needs to be higher. Put differently, I don't see why one assumes that the peak expression disallows any important function in a non-peak period.

R6: We kindly disagree with the referee on the relevance of our observations on hiPSCs. While it is true that the peak of expression of a TF will not necessarily correlate with its regulatory role, a similar clustering of key TFs according to their expression profile in different species (as we show in Figures 3 and 8) strongly argues for

a conserved cis-regulatory logic that we hope to explore in further studies. In addition, sampling of hiPSCs-RPE cultures every week is perfectly in agreement with the differentiation pace observed in these cultures, which acquire pigmentation after a month. This is an important point as it validates some of our results in a different vertebrate system, and may be useful to improve hiPSCs-to-RPE differentiation protocols. For these reasons, we believe that the hiPSCs data should be maintained in the paper.

REVIEWERS' COMMENTS

Reviewer #1 (Remarks to the Author):

After reviewing the revised manuscript and author response, my opinion is that the authors addressed my comments in a satisfying manner.

Reviewer #2 (Remarks to the Author):

The study by Buono et al. investigates the gene networks controlling the bifurcation of two lineages of the ocular neuroectoderm in the embryonic Zebrafish eye: retinal pigmented epithelium and neural retina. The study presents a global, unbiased analysis of differences in the transcriptomic and regulatory regions of the neural and pigmented ocular lineages, and a phenotypic screen of a selected set of identified targets expected to play a role in this specification process. The results indicate redundancy of transcription regulators, the differences in the lineages' genomic architectures, and the importance of YAP/TAZ in the regulation of genes important for cell adhesion and morphology.

In the revised version, the authors have addressed most of the reviewers' comments and queries. The manuscript is well written and the data are clearly presented. A main concern of all of the reviewers was the validity of the CRISPR screen and the need to substantiate, using a comprehensive functional study, one of the conclusions from the genomic analyses. The authors substantially improved the CRISPR-CAS9 F0 screen by adding histological analyses, quantifications of eye size and analysis of a later developmental stage. However, the detailed functional analysis of one of their predictions remains unresolved and is a topic for future studies. The manuscript has been substantially improved and is now better suited for publication in Nature Communications.

One additional comment: Add a table of the genes targeted by the CRISPR screen and include the GO terms, expression profiles, cis regulatory element composition/conservation/dynamics, number of paralogues and the outcome of the screen.

Reviewer #3 (Remarks to the Author):

I have re-reviewed the manuscript by Buono et al. In general, I am satisfied with their responses to the previous review and remain positive about the manuscript. Indeed, I think it makes a nice contribution to the field conceptually, in elucidating the molecular underpinnings of the bifurcation of the retina and RPE lineages. The datasets will also be of use to many in the field.

My main concern previously was in the F0 CRISPR screen and I think the additional controls and data added by the authors in revision have mitigated this concern.

I am still a bit concerned about the bhlh40e transgenic and its RPE specificity. It is expressed in non-RPE cells. Thus, I would think the authors must indicate that RPE was enriched, but that there is likely some contamination by CMZ and neural crest. I think the data here are useful but this caveat should be noted.

Finally, I still think the hiPSC data are forced, do not add much to the story and, in fact, detract a bit from the nice studies up to that point. In their rebuttal, the authors did not make a compelling argument, in my opinion, as to why these should remain. That said, I leave this to the Editor and authors to hash out. The concern raised previously about interpretation has been adequately addressed, however, so this criticism is all stylistic.

REVIEWERS' COMMENTS

Reviewer #1 (Remarks to the Author):

After reviewing the revised manuscript and author response, my opinion is that the authors addressed my comments in a satisfying manner.

R1: We thank this reviewer for all the critical comments, which have certainly help to improve the work.

Reviewer #2 (Remarks to the Author):

The study by Buono et al. investigates the gene networks controlling the bifurcation of two lineages of the ocular neuroectoderm in the embryonic Zebrafish eye: retinal pigmented epithelium and neural retina. The study presents a global, unbiased analysis of differences in the transcriptomic and regulatory regions of the neural and pigmented ocular lineages, and a phenotypic screen of a selected set of identified targets expected to play a role in this specification process. The results indicate redundancy of transcription regulators, the differences in the lineages' genomic architectures, and the importance of YAP/TAZ in the regulation of genes important for cell adhesion and morphology.

In the revised version, the authors have addressed most of the reviewers' comments and queries. The manuscript is well written and the data are clearly presented. A main concern of all of the reviewers was the validity of the CRISPR screen and the need to substantiate, using a comprehensive functional study, one of the conclusions from the genomic analyses. The authors substantially improved the CRISPR-CAS9 F0 screen by adding histological analyses, quantifications of eye size and analysis of a later developmental stage. However, the detailed functional analysis of one of their predictions remains unresolved and is a topic for future studies. The manuscript has been substantially improved and is now better suited for publication in Nature Communications.

R1: We thank this reviewer for the critical comments and suggestions, and appreciate the positive words on the revised version of the manuscript.

One additional comment: Add a table of the genes targeted by the CRISPR screen and include the GO terms, expression profiles, cis regulatory element composition/conservation/dynamics, number of paralogues and the outcome of the screen.

R2: We are now including all this information in Dataset S12.

Reviewer #3 (Remarks to the Author):

I have re-reviewed the manuscript by Buono et al. In general, I am satisfied with their responses to the previous review and remain positive about the manuscript. Indeed, I think it makes a nice contribution to the field conceptually, in elucidating the molecular underpinnings of the bifurcation of the retina and RPE lineages. The datasets will also be of use to many in the field. My main concern previously was in the F0 CRISPR screen and I think the additional controls and data added by the authors in revision have mitigated this concern.

R1: We thank this reviewer for the helpful comments and suggestions and his/her appreciation of the potential impact of the work.

I am still a bit concerned about the bhlh40e transgenic and its RPE specificity. It is expressed in non-RPE cells. Thus, I would think the authors must indicate that RPE was enriched, but that there is likely some contamination by CMZ and neural crest. I think the data here are useful but this caveat should be noted.

R2: As already stated the bhlh40 line used is specific of the RPE precursors at the early stages considered in our study. It is only later in development, after the complete folding of the OC at 24 hpf, that GFP expression is

detected in the ciliary marginal zone (CMZ) and very few neural crest cells surrounding the eye. This is clearly shown in our paper Moreno-Marmol et al. 2020, BioRxiv (doi.org/10.1101/2020.09.23.310631) and in Figure S12 in this work. We are confident that even at the later stage analysed (23 hpf) there is only a very minor contamination of CMZ and neural crest cells in the isolated RPE population. However, in line with the reviewer's suggestion, we have included the following comment in the discussion:

"Although expression driven by the RPE-specific element E1_bHLHe40 has also been reported in the ciliary marginal zone and periocular neural crest cells, this is only reported after optic cup folding (Moreno-Marmol et al. 2020). Therefore, only a minor contamination, if any, would be expected in RPE precursors isolated at 23 hpf."

Finally, I still think the hiPSC data are forced, do not add much to the story and, in fact, detract a bit from the nice studies up to that point. In their rebuttal, the authors did not make a compelling argument, in my opinion, as to why these should remain. That said, I leave this to the Editor and authors to hash out. The concern raised previously about interpretation has been adequately addressed, however, so this criticism is all stylistic.

R3: We again disagree with the referee on the relevance of our observations on hiPSCs. We still think that the expression data in human cells validates some of our results in a different vertebrate species, and may be useful to improve hiPSCs-to-RPE differentiation protocols. Furthermore, we think that removing this information may reduce the visibility of our study, which may serve as a reference for future and more comprehensive studies in human cells.